# OOD-Chameleon:
# Is Algorithm Selection for OOD Generalization Learnable?

**Liangze Jiang** [1 2]  **Damien Teney** [2]

## Abstract

Out-of-distribution (OOD) generalization is challenging because distribution shifts come in many forms. Numerous algorithms exist to address specific settings, but *choosing the right training algorithm for the right dataset* without trial and error is difficult. Indeed, real-world applications often involve multiple types and combinations of shifts that are hard to analyze theoretically.

**Method.** This work explores the possibility of *learning* the selection of a training algorithm for OOD generalization. We propose a proof of concept (OOD-Chameleon) that formulates the selection as a multi-label classification over candidate algorithms, trained on a *dataset of datasets* representing a variety of shifts. We evaluate the ability of OOD-Chameleon to rank algorithms on unseen shifts and datasets based only on dataset characteristics, i.e. without training models first, unlike traditional model selection.

**Findings.** Extensive experiments show that the learned selector identifies high-performing algorithms across synthetic, vision, and language tasks. Further inspection shows that it learns non-trivial decision rules, which provide new insights into the applicability of existing algorithms. Overall, this new approach opens the possibility of better exploiting and understanding the plethora of existing algorithms for OOD generalization. Code: https://github.com/LiangzeJiang/OOD-Chameleon

## 1. Introduction

Out-of-distribution (OOD) generalization refers to a model's ability to remain accurate when the distributions of training

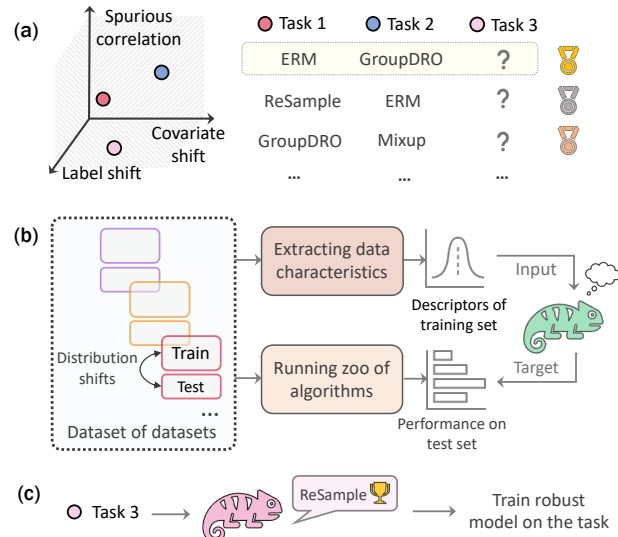

*Figure 1.* **Choosing the right training algorithm is an often-overlooked key factor in OOD generalization.** **(a)** Different training algorithms perform differently on different distribution shifts. **(b)** We propose to automate the selection of a learning algorithm. We train a selector (OOD-Chameleon) on a "dataset of datasets" that exemplifies a variety of shifts. **(c)** For a novel task/dataset, the learned selector predicts the best algorithm to train a robust model.

and test data differ. "OOD" is a catch-all term that encompasses many types of distribution shifts (Wiles et al., 2021; Ye et al., 2022; Nagarajan et al., 2020). In medical imaging for example (Oakden-Rayner et al., 2020), a model may process scans from various demographics (*covariate shift*), pathologies (*label shift*), and co-occurrences of patient attributes (*spurious correlations*). In many applications, these shifts are combined, making a theoretical analysis challenging (Wiles et al., 2021; Yang et al., 2023; Jeon et al., 2025).

**The trade-offs of learning algorithms.** There exists a multitude of algorithms[1] designed to improve OOD generalization, from standard ERM (Vapnik, 2000) to simple interventions such as Resampling (Japkowicz & Stephen,

---

[1]EPFL, Switzerland [2]Idiap Research Institute, Switzerland. Correspondence to: Liangze Jiang <liangze.jiang@epfl.ch>.

*Proceedings of the 42nd International Conference on Machine Learning*, Vancouver, Canada. PMLR 267, 2025. Copyright 2025 by the author(s).

---

[1]In this work, "algorithm" refers to a method, or particular element thereof that takes the training data and produces a trained model.

2002), GroupDRO (Sagawa et al., 2019), and much more complex ones (Liu et al., 2023a). However, each algorithm usually targets specific distribution shift settings (Figure 1a). Indeed, numerous studies by Gulrajani & Lopez-Paz (2020) and others (Wiles et al., 2021; Nguyen et al., 2021; Ye et al., 2022; Liang & Zou, 2022; Benoit et al., 2023; Yang et al., 2023) showed that no single intervention surpasses ERM across a range of datasets. The reason is that OOD generalization is fundamentally underspecified (D'Amour et al., 2022; Teney et al., 2022) and different methods make different assumptions that can each shine in different conditions. This means that **choosing the right learning algorithm for a given situation is important to improve OOD generalization**. This possibility has thus been tantalizing:

> *"It would be helpful for practitioners to be able to select the best algorithms without comprehensive evaluations and comparisons." (Wiles et al., 2021)*

However, identifying the right algorithm without trial and error is challenging due to the intricate *interplay* between the algorithms' inductive biases and the complex types of shifts in real-world data. Moreover, the trial-and-error might be impossible because of the lack of OOD evaluation data.

**This paper proposes to *learn* to predict, given a dataset, the best learning algorithm for training a robust model.** This *a priori* prediction contrasts with traditional model selection that first requires training *many* models or relies on restrictive heuristics (e.g. accuracy/agreement on the line or activation coverage, see Garg et al. (2022); Baek et al. (2022); Miller et al. (2021); Teney et al. (2023); Liu et al. (2023b)). Concurrent work by Bell et al. (2024) selects algorithms for spurious correlations only, based on past performance on benchmarks most similar to the new task. We similarly rely on past performance, but (i) we do not require any training run for the new tasks, (ii) we address more types of shifts and (iii) we use non-linear learnable predictors instead of non-parametric nearest-neighbor retrieval.

**OOD-Chameleon.** We frame the algorithm selection as a *multi-label classification* over a set of representative algorithms.[2] We train our model to predict the algorithms' suitability given some dataset descriptor (Figure 1b), such as dataset size, degrees of shifts, etc. As training examples, we build a *dataset of datasets* representing diverse types, magnitudes, and combinations of shifts, by re-sampling existing datasets with fine-grained annotations, such as CelebA (Liu et al., 2015) or CivilComments (Borkan et al., 2019)). The

model thus learns to exploit the algorithms' performance in a variety of conditions. It can then recommend an appropriate algorithm for any new task (Figure 1c).

**Results.** We evaluate the model on seven applications in **synthetic** (Sagawa et al., 2020) , **vision** (CelebA, MetaShift, OfficeHome, etc.) and **language** domains (CivilComments, MultiNLI). The model is capable of selecting suitable algorithms for unseen shifts and datasets, achieving significantly lower test error than any static choice of algorithm. We verify that it achieves this by learning non-trivial, non-linear relations between dataset characteristics and algorithms' performance. Crucially, we examine the learned model to extract its decision rules, which reveal which dataset characteristics are important for various algorithms to outperform one another.

**Contributions.** Our findings support a positive answer to the title: OOD generalization can be improved by learning to better use existing algorithms. This helps in dealing with complex types of shifts that are difficult to study theoretically, and provides insights on the applicability of existing algorithms. Our contributions are summarized as follows.

- We are the first to identify and formalize the problem of **algorithm selection for OOD generalization** (§2).

- We present a solution (OOD-Chameleon) that takes in dataset characteristics and predicts the suitability of candidate algorithms to train a robust model (§2.2).

- We show with extensive experiments that OOD-Chameleon (i) adaptively chooses suitable algorithms (§4), (ii) learns non-trivial data/algorithm interactions (§4.5) that transfer to unseen shifts and datasets (§4.3–4.4), (iii) reveals which dataset properties make algorithms outperform one another (§4.5).

- We release a tool for future research (§3) to construct datasets with controlled and potentially mixed shift types, and magnitudes, including spurious correlations, label shifts and covariate shifts.

## 2. OOD-Chameleon: Algorithm Selection for OOD Generalization

### 2.1. Is the Selection of the Best Algorithm Possible?

The no-free-lunch theorem (Wolpert & Macready, 1997) prevents a universal solution to the selection of the best learning algorithm. But distribution shifts appearing in real-world data are not arbitrary[3] (Wiles et al., 2021; Goldblum et al., 2024). A learning approach could thus be trained for effective algorithm selection on a particular *distribution*

---

[2]We consider algorithms with proven efficacy on different types of shifts that are actually used in ML deployments, and not only in one-off academic papers (see discussion in Section 4): ERM, GroupDRO, Oversample, Undersample, and Logits adjustment.

[3]If shifts were arbitrary, inductive reasoning and machine learning would not be possible (Wolpert, 1996).

*of distribution shifts*. Moreover, measurable properties of a dataset can be indicative of an imbalance/shift, and thus of the suitability of algorithms. We therefore propose to train a selector that takes dataset characteristics as input (e.g. dataset size, degree of distribution shifts), and predicts the suitability of various candidate algorithms.

**Conjecture 1 (Learnability).** *There exists a learnable mapping from measurable dataset characteristics (e.g., size, feature imbalance, shift indicators) to algorithm suitability, such that a selector trained on a sufficiently diverse and densely sampled distribution of distribution shifts can generalize to unseen, but related, shifts **within** the support of the training distribution.*

This assumption is crucial since there is no guarantee (just like any other learning method) for the selector to generalize beyond the support of the training distribution of distribution shifts. Yet, as we will present, our approach encompasses a wide variety of shift types and magnitudes, which can be densely sampled as training data for the selector.

**Distribution shift taxonomy.** We consider the common distribution shift taxonomy based on three types (Yang et al., 2023): **covariate shifts** (CS), **label shifts** (LS), and **spurious correlations** (SC). The first two correspond to changes in $P(X)$ and $P(Y)$ where $X$ and $Y$ represent inputs and class labels. To formalize spurious features, we view $X$ as containing core features $X_c$ always predictive of $Y$, and possibly other features $X_a$ whose presence is also indicated by an attribute $A$. A spurious correlation means that $A$ and $X_a$ are correlated with $Y$ in the training data, even though $X_a$ cannot be relied on to make correct predictions on test data. This implies therefore a shift of $P(Y|X)$.

To measure OOD performance, we use standard **worst-group** (WG) **error** on test data, where **groups** $\mathcal{G} \in Y \times A$ are class–attribute combinations (Sagawa et al., 2019). WG performance is also what we will (indirectly) optimize for. The rationale is that it is independent of the test distribution and can thus be addressed by considering only the distribution (imbalances) of *training* data. WG performance thus promotes general robustness to distribution shifts.

We focus primarily on the most common setting in OOD generalization (Yong et al., 2022) where the training data includes labels of a potentially spurious attribute $A$. We also evaluate the use of pseudo attributes in Appendix F.

### 2.2. Algorithm Selection as a Supervised Learning Task

Our eventual goal is to train robust models on future (yet unknown) tasks. A **task** is defined by the training and test splits that make up a dataset $D = D^{\mathrm{tr}} \cup D^{\mathrm{te}} = \{(x_i, y_i)\}_{i=1}^{n} \cup \{(x_i, y_i)\}_{i=1}^{n_{\mathrm{te}}}$. Training a model means running a learning algorithm $\mathcal{A}(\cdot)$ to obtain a parametrized

model: $\mathcal{A}(D^{\mathrm{tr}}) = h_\theta$. The OOD performance of $\mathcal{A}$ is defined as the WG error of $h_\theta$ on $D^{\mathrm{te}}$.

We propose OOD-Chameleon to automate the choice of the best algorithm among $M$ candidates for unseen tasks. This is motivated by the many existing results showing that different algorithms perform differently in different conditions (see Section 1). OOD-Chameleon will learn to choose among candidate algorithms based on their past performance in a variety of conditions. To do so, we build a **meta-dataset** $\mathbb{D}$ as the training data, which is a dataset of datasets representing a variety of distribution shifts (Section 3). Each $j$th dataset is associated with an algorithm $\mathcal{A}_m$ and its WG performance $P_{jm} \in [0, 1]$ on the dataset. Formally, the meta-dataset is defined as a collection of triplets:

$$\mathbb{D} = \{ f(D_j^{\mathrm{tr}}), \ \mathcal{A}_m, \ P_{jm} \} \tag{1}$$

where $f(D_j^{\mathrm{tr}})$ is a **dataset descriptor** (Rivolli et al., 2022).

**Dataset descriptors.** To make the learning tractable, we need a function $f : \mathrm{Supp}(D^{\mathrm{tr}}) \to \mathbb{R}^l$ that summarizes various dataset properties in a fixed-length vector. Recent work examined properties relevant to the performance of learning algorithms (Nagarajan et al., 2020; Hermann et al., 2023; Yang et al., 2024; Chen et al., 2022; Ye et al., 2022; Wang et al., 2024) but most cannot be measured without first training a model, defeating our purpose. Others (Arango et al., 2023; Öztürk et al., 2022) used trivial properties (e.g. number of classes) that are clearly insufficient in our case.

We consider two sets of properties for our dataset descriptors: (i) **distribution shift characteristics** and (ii) **data complexity characteristics**. The former set includes the shift magnitudes defined in Section 2.1 ($d_{\mathrm{sc}}, d_{\mathrm{ls}}, d_{\mathrm{cs}}$) and the availability of the spurious feature ($r$).[4] These are provided as ground truth or estimated (see Appendix F). The latter set includes the size of the training set $n$ and the input dimensionality $d$. We leave for future work the possibility of learning descriptors end-to-end with the algorithm selection.

**Training an algorithm selector.** We can now use $\mathbb{D}$ (Eq. 1) to train an algorithm selector (typically a neural network) that predicts, given the training set of a task, the most suitable algorithm. We can formulate this concept either as a regression or a classification. To implement the selector as a **regression**, we train a model $\phi(w, \cdot) : f(D^{\mathrm{tr}}) \times \mathcal{A} \to \mathbb{R}$ that maps a dataset descriptor $f(D^{\mathrm{tr}})$ and algorithm identifier $\mathcal{A}$[5] to a predicted performance. We could train it for the regression objective:

$$\min_w \mathbb{E}_{\mathbb{D}} \, \mathcal{L}_{\mathrm{MSE}} \Big( \phi\big(w, \{f(D_j^{\mathrm{tr}}), \mathcal{A}_m\}\big), P_{jm} \Big) \tag{2}$$

---

[4]Similar concepts include signal/noise ratio (Yang et al., 2024), magnitude (Wang et al., 2024; Joshi et al., 2023), simplicity (Qiu et al., 2024), spurious/core information ratio (Sagawa et al., 2020).
[5]By abuse of notation, $\mathcal{A}$ is an algorithm's 1-hot identifier here.

where $\mathcal{L}_{\mathrm{MSE}}$ is the mean square error. The selector can then predict the performance of several algorithms on an unseen task. The top prediction is used to train a robust model.

The alternative implementation as a **multi-label classification task** is motivated by the fact that neural networks are easier to train for classification than regression (Devroye et al., 2013). A classification also aligns better with the goal of selecting algorithms rather than estimating their absolute performance. We choose a multi-label formulation rather than a single-label/multi-class because *the algorithms do not compete against one another*, and multiple algorithms can sometimes be equally suitable.

Specifically, we define the training classification labels as follows. For $j$th dataset in $\mathbb{D}$, we have $M$ (the number of algorithms) records $\{f(D_j^{\mathrm{tr}}), \mathcal{A}_m, P_{jm}\}_{m=1}^M$. We aggregate each such set of $M$ records into a *single* training sample $\{f(D_j^{\mathrm{tr}}), Y_{\mathcal{A}}\}$ where $Y_{\mathcal{A}} \in \{0,1\}^M$ is a one- or multi-hot vector indicating the suitability of the candidate algorithms. An algorithm is considered suitable if $(P_{jm} - \min_m P_{jm}) \leq \epsilon$ for a small threshold $\epsilon$ (we use $0.05$ for all experiments and perform ablation study in Appendix E.2). This aggregation converts performance numbers into discrete labels, which acts as a sort of "denoising" since algorithms with close performance are deemed similarly suitable. We use these labels to train a multi-label classifier $\phi(w, \cdot) : f(D^{\mathrm{tr}}) \to \{0,1\}^M$ for the objective:

$$\min_w \; \mathbb{E}_{\mathbb{D}} \; \mathcal{L}_{\mathrm{BCE}}\Big(\phi\big(w, f(D_j^{\mathrm{tr}})\big), Y_{\mathcal{A}}\Big) \qquad (3)$$

where $\mathcal{L}_{\mathrm{BCE}}$ is the binary cross-entropy. Unless noted, OOD-Chameleon refers to this multi-label classification implementation. We use the simplest objective for clarity and proof-of-concept; future work can adjust the objective for desirable properties, such as a DRO-like objective (Rahimian & Mehrotra, 2022) or a multi-task objective.

**Applying the algorithm selector.** The trained selector can then predict the suitability of algorithms on an unseen task. When multiple algorithms are predicted as suitable at test-time, the one with the *top prediction logit* (which corresponds to the most confident one) is used to train a robust model, see Appendix E.3 for an ablation study on test-time algorithm selection strategies.

### 2.3. Discussion and Summary

OOD-Chameleon can be seen as a data-driven performance prediction, or a learned *a priori* model selection (i.e. before training any target model). A related line of work is the data-driven selection among pre-trained vision models (Zhang et al., 2023; Achille et al., 2019; Öztürk et al., 2022). In all cases, the intuition is to exploit knowledge of **past performance in known conditions** for new tasks. In our case, (i) due to the supervised classification formula-

tion, classical results from statistical learning theory apply to the algorithm selector , and (ii) the data-driven view is particularly valuable as **it handles complex shifts that are difficult to analyze theoretically**. For example, a label shift is effectively addressed with class balancing (Idrissi et al., 2022). It is much less clear, however, how to address a label shift combined with a covariate shift and a mild spurious correlation, for example Garg et al. (2023).

---

**In summary**, building OOD-Chameleon proceeds in three steps (see also Figure 1b).

1. **Obtaining a collection of datasets** with a variety of distribution shifts (see Section 3).
2. **Assembling the meta-dataset** ($\mathbb{D}$), i.e. pre-computing dataset descriptors and training models with the candidate algorithms to obtain their "ground truth" performance.
3. **Training the algorithm selector** ($\phi$) on $\mathbb{D}$.

---

## 3. A Tool to Construct Distribution Shifts

We now describe the construction of the dataset collection, which exemplifies various shift types and magnitudes. The same tool can be used for future research to obtain datasets with controlled, mixed-type shifts. It takes as input a dataset that have fine-grained annotations (i.e. each sample is annotated with its class label and one or more attributes), such as CelebA (Liu et al., 2015) or CivilComments (Borkan et al., 2019), and outputs a dataset with controlled distribution shifts by resampling the source dataset.

**Quantifying the distribution shifts.** To have precise control over the degree of shifts, we need an unambiguous way to quantify them. Assuming the test set is balanced (i.e. all groups are equally represented), then how imbalanced/biased the training data is (to test data) directly reflects the degree of shifts. Therefore, we define the following degrees:

- **Spurious correlation** ($d_{\mathrm{sc}}$): The ratio of training samples whose class label and attribute agree, i.e. where a correct classification of the attribute entails a correct classification of the class.

- **Label shift** ($d_{\mathrm{ls}}$): The imbalanceness of training class label distribution.

- **Covariate shift** ($d_{\mathrm{cs}}$): The imbalanceness of training attribute distribution.

See Figure 2 for an illustrative example with a 2-way classification of shapes with two color attributes, where $|\cdot|$ is the set cardinality, and $\sum_i |\mathcal{G}_i| = n$ the size of training set $D^{\mathrm{tr}}$, with $|\mathcal{G}_i|$ being the number of samples in each group. These degrees are in $[0, 1]$ by definition and $0.5$ means the absence

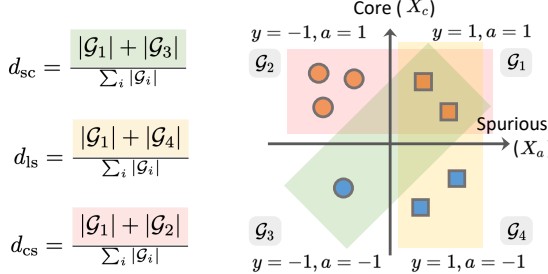

$$d_{\text{sc}} = \frac{|\mathcal{G}_1| + |\mathcal{G}_3|}{\sum_i |\mathcal{G}_i|}$$

$$d_{\text{ls}} = \frac{|\mathcal{G}_1| + |\mathcal{G}_4|}{\sum_i |\mathcal{G}_i|}$$

$$d_{\text{cs}} = \frac{|\mathcal{G}_1| + |\mathcal{G}_2|}{\sum_i |\mathcal{G}_i|}$$

*Figure 2.* Illustration of the quantification of distribution shifts. In this example, $d_{\text{sc}} = 3/8$, $d_{\text{ls}} = 0.5$, and $d_{\text{cs}} = 5/8$.

of a shift. In practice, we can use different class-attribute pairs available in datasets such as CelebA or CivilComments (see Table 9 for some pairs). Different class-attribute pairs naturally induce different availabilities of spurious features, depending on how the attributes are embedded in the data.

**Constructing datasets.** Next, given a desired training set size $n$ and a triple $(d_{\text{cs}}, d_{\text{ls}}, d_{\text{sc}}) \in [0, 1]^3$ that indicates a desired degree of shifts, we could solve linear equations (the ones in Figure 2, plus $\sum_i |\mathcal{G}_i| = n$) to obtain the necessary number of samples per group $\mathcal{G}_i$ to achieve these properties in the training set. The test set $D^{\text{te}}$ is balanced, i.e. $|\mathcal{G}_i| = n_{\text{te}}/$num. of groups. We sample the corresponding number of samples from the source dataset and compose them into a task with desired properties. We then perform this process many times with different properties to obtain many different datasets. See Appendix B for details and Figure 7 for example datasets.

## 4. Experiments

We evaluate OOD-Chameleon on seven applications from three domains: **synthetic** (Sagawa et al., 2020), **vision** (CelebA (Liu et al., 2015), MetaShift (Liang & Zou, 2022), OfficeHome (Venkateswara et al., 2017), Colored-MNIST (Arjovsky et al., 2020), and **language** (CivilComments (Borkan et al., 2019), MultiNLI (Williams et al., 2017)). See details of the datasets in Appendix A. As we will show, all cases benefit from the adaptive selection. We also verify that it achieves this by learning non-trivial decision rules that are transferrable across tasks. Finally, we examine the learned selector to understand when the algorithms can outperform one another.

### 4.1. Experimental Setup

For each domain (synthetic, vision, language), we create a meta-dataset $\mathbb{D}$ then train an algorithm selector. See Appendices C–D for details. We then evaluate the generalizability of the algorithm selector on unseen tasks with properties disjoint from $\mathbb{D}$. We report the 0–1 accuracy (higher is better), which considers an algorithm prediction as correct if

it is in the set of "suitable" algorithms (as defined in Section 2.2). We also report the worst-group error (lower is better) of models trained with the selected algorithms, averaged across unseen tasks. The former evaluates the algorithm selection itself. The latter evaluates the actual benefits in error reduction by relying on the algorithm selector.

**Candidate algorithms.** We select five algorithms, namely *ERM* (Vapnik, 2000), *GroupDRO* (Sagawa et al., 2019), *oversampling* minority groups (Japkowicz & Stephen, 2002), *undersampling* majority groups, and *logits adjustment* (Menon et al. (2021); Nguyen et al. (2021); Kini et al. (2021), which encourages a relative larger margin for the minority groups. We choose to focus on algorithms (i) with strong proven performance, competitive or superior to more complex ones (Nguyen et al., 2021; Gulrajani & Lopez-Paz, 2020; Idrissi et al., 2022; Yang et al., 2023), (ii) that do not require extensive hyperparameter tuning, (iii) that each address different types of shifts (Nguyen et al., 2021), (iv) that clearly belong to different families, namely regularization-, reweighting-, and margin-based approaches. Building OOD-Chameleon on other existing algorithms is a direct extension that however requires more computational resources, which we thus leave for future work.

**Algorithm selection baselines and ablations.**
- **Random selection**: randomly selecting algorithms.
- **Global best**: choosing the single best algorithm according to its performance on all tasks in the meta-dataset.
- **Naive descriptors**: using trivial dataset properties from Öztürk et al. (2022) as input to train the selector, instead of our dataset descriptors. This evaluates if our descriptors provide relevant information.
- **Oracle selection**: upper bound, which uses the best algorithm for each dataset.
- **Regression**: selector trained from the regression (Eq. 2).

**Models.** The algorithm selector is implemented as an MLP unless otherwise noted. We evaluate other implementations in Table 5. As for target models, the synthetic experiments (Section 4.2) follow Sagawa et al. (2020) and use a linear model, which is sufficient to solve the synthetic task. The vision experiments (Section 4.3) use linear probing on a pre-trained ResNet18 (He et al., 2015) or CLIP model (ViT-B/32) (Radford et al., 2021). We show in Appendix E.1 that OOD-Chameleon is effective for both linear probing and fine-tuning paradigms. The language experiments (Section 4.4) use linear probing on a pre-trained BERT (Devlin et al., 2019) or a BERT fine-tuned on CivilComments. In all cases, we train for long enough to ensure convergence with identical hyperparameters across runs. The rationale is that hyperparameter search should not be allowed since OOD validation data cannot be relied on (otherwise it could be simply used as training data to achieve OOD generalization).

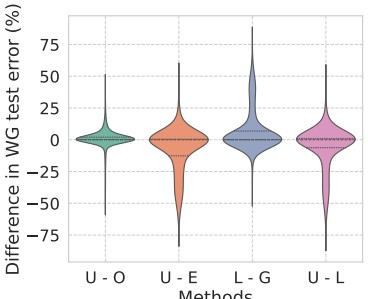 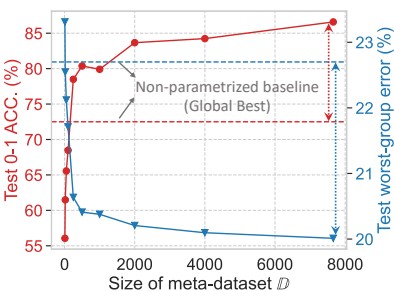 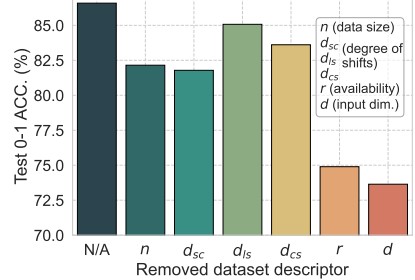

*Figure 3.* (**Left**) Discrepancy in performance among algorithms. Across many tasks, we compare pairs of algorithms (e.g. L-G means comparing Logits adjustment and GroupDRO) and show the distribution of performance differences in worst-group test error. (**Middle**) The generalizability of the algorithm selector improves with a larger meta-dataset. (**Right**) We estimate the importance of each piece of the dataset descriptor with leave-one-descriptor-out training of the algorithm selector.

## 4.2. Synthetic Experiments

**Data.** We consider a binary classification following closely the synthetic example from Sagawa et al. (2020). The group distribution is implicitly defined from the definition of inputs as $x = [x_c, x_a] \in \mathbb{R}^{2d}$ with $x_c$ and $x_a$ of dimension $d$ generated from Gaussian distributions:

$$x_c \mid y \sim \mathcal{N}\left(y\mathbf{1}, \sigma_c^2 I_d\right), \quad x_a \mid y \sim \mathcal{N}\left(a\mathbf{1}, \sigma_a^2 I_d\right).$$

The availability of the spurious features is defined as $r = \sigma_c^2/\sigma_a^2$. We create $\sim 7{,}000$ tasks for $\mathbb{D}$ spanning different $d_{sc}, d_{ls}, d_{cs}$, training sizes $n$, dimensionality $d$, availability $r$, and another $\sim 2{,}000$ for evaluation (details in Appendix C).

**Results.** Table 1 first confirms, from the gap between random selection and oracle selection, that the ability to choose the right algorithm provides substantial benefits. We show that OOD-Chameleon is accurate at predicting the most suitable algorithm on unseen tasks. Both the 0–1 accuracy and the worst-group error are substantially better than the baselines.

Additional results in Figure 3 (left) indeed show that different algorithms perform differently across tasks/datasets. Figure 3 (middle) shows that OOD-Chameleon is more accurate with the meta-dataset scaled up. Yet, it already significantly outperforms the best non-parametrized baseline ("Global best") with a relatively small ($\sim 200$) meta-dataset. In Figure 3 (right), we conduct a leave-one-descriptor-out training of OOD-Chameleon, excluding one element from the descriptor at a time. The drops in accuracy (compared to the leftmost bar) show that every element provides useful information. Those that matter most are the input dimensionality $d$, then the availability $r$ and degree $d_{sc}$ of the spurious correlation.

### 4.3. Vision Experiments

**Data.** We generate 720 tasks from CelebA (Liu et al., 2015), again with various data sizes, types of shifts, spurious

*Table 1.* **Results on the synthetic tasks.** The learned algorithm selector approaches the performance of the oracle selection.

| Methods | 0–1 ACC. (%) ↑ | WG error (%) ↓ | Remarks |
|---|---|---|---|
| Oracle selection | 100 | 19.0 | Upper bound |
| Random selection | $62.9_{\pm 0.6}$ | $24.0_{\pm 0.1}$ | Non-parametrized |
| Global best | $72.5_{\pm 0.7}$ | $22.7_{\pm 0.1}$ | Non-parametrized |
| Naive descriptors | $52.1_{\pm 0.1}$ | $23.9_{\pm 0.2}$ | Öztürk et al. (2022) |
| Regression | $79.7_{\pm 0.7}$ | $20.4_{\pm 0.3}$ | Equation 2 |
| OOD-Chameleon | $\mathbf{86.3}_{\pm 0.4}$ | $\mathbf{19.9}_{\pm 0.1}$ | Equation 3 |

features, etc. We simulate different availabilities of the spurious feature by using different attribute annotations from CelebA, e.g. `mouth slightly open` as class label and `wearing lipstick` as spurious attribute (details in Appendix D). With MetaShift, we similarly use {`cat`, `dog`} as classes and {`indoor`, `outdoor`} as attributes to generate 129 tasks. With OfficeHome, we created 100 tasks by randomly sampling a pair of domains and a pair of classes for each dataset (e.g. classifying {`pen`, `knife`} in {`Art`, `Clipart`}). For each task of OfficeHome, the number of samples of each group is naturally imbalanced, creating different distribution shifts by design. With Colored-MNIST, we use the digits as classes and colors as attributes to generate 180 tasks. We build the meta-dataset with $80\%$ of the CelebA tasks to train the selector, then evaluate it either on the remaining $20\%$ of CelebA tasks, or on the MetaShift, OfficeHome, Colored-MNIST tasks, respectively.

**The availability of spurious features.** Unlike with synthetic data, there is no straightforward measure of the availability $r$ for real-world data. We use a proxy $r = \sum_y l_y / \sum_a l_a$ where $l_y$ (resp. $l_a$) is the average distance between every sample and the center of the cluster of its class (resp. attribute). Distances are measured in the embedding space of the corresponding pre-trained model (Section 4.1). Intuitively, the availability is higher when the cluster w.r.t. the attribute is more compact than to the label (see details and discussions in Appendix D.1).

*Table 2.* **Results on vision tasks**. The selector is trained on a meta-dataset built from CelebA, and then evaluated on a separate set of tasks from CelebA (Left) as well as MetaShift (Right). It remains accurate across datasets, meaning that it learns generalizable decision rules.

| Methods | CelebA | | | | MetaShift | | | |
|---|---|---|---|---|---|---|---|---|
| | ResNet18 | | CLIP (ViT-B/32) | | ResNet18 | | CLIP (ViT-B/32) | |
| | 0–1 ACC. ↑ | WG error ↓ | 0–1 ACC. ↑ | WG error ↓ | 0–1 ACC. ↑ | WG error ↓ | 0–1 ACC. ↑ | WG error ↓ |
| Oracle selection | 100 | 44.9 $_{\pm 0.2}$ | 100 | 36.3 $_{\pm 0.3}$ | 100 | 36.4 $_{\pm 0.3}$ | 100 | 24.5 $_{\pm 0.3}$ |
| Random selection | 28.5 $_{\pm 0.5}$ | 53.4 $_{\pm 0.6}$ | 27.0 $_{\pm 0.3}$ | 45.4 $_{\pm 0.2}$ | 33.3 $_{\pm 0.8}$ | 43.1 $_{\pm 0.2}$ | 34.1 $_{\pm 0.3}$ | 32.3 $_{\pm 0.1}$ |
| Global best | 35.7 $_{\pm 1.0}$ | 51.3 $_{\pm 0.5}$ | 32.6 $_{\pm 0.4}$ | 43.4 $_{\pm 0.1}$ | 39.4 $_{\pm 0.3}$ | 42.4 $_{\pm 0.2}$ | 36.4 $_{\pm 0.3}$ | 31.7 $_{\pm 0.4}$ |
| Naive descriptors | 64.5 $_{\pm 0.9}$ | 49.7 $_{\pm 0.5}$ | 66.8 $_{\pm 0.6}$ | 40.7 $_{\pm 0.4}$ | 68.8 $_{\pm 0.7}$ | 40.3 $_{\pm 0.3}$ | 66.6 $_{\pm 0.3}$ | 28.1 $_{\pm 0.4}$ |
| Regression | 53.6 $_{\pm 1.1}$ | 49.8 $_{\pm 0.5}$ | 46.7 $_{\pm 0.9}$ | 41.2 $_{\pm 0.3}$ | 51.2 $_{\pm 0.3}$ | 40.9 $_{\pm 0.2}$ | 49.2 $_{\pm 0.3}$ | 28.3 $_{\pm 0.2}$ |
| OOD-Chameleon | **75.0** $_{\pm 1.3}$ | **47.7** $_{\pm 0.2}$ | **78.5** $_{\pm 0.8}$ | **39.1** $_{\pm 0.2}$ | **80.6** $_{\pm 0.7}$ | **39.0** $_{\pm 0.3}$ | **76.7** $_{\pm 0.5}$ | **27.2** $_{\pm 0.3}$ |

*Table 3.* **Comparison with static algorithm selection**, i.e. using the same one used on all test tasks. OOD-Chameleon performs much better by adaptively choosing an algorithm for each task. It approaches the performance of an oracle selection, as well as its distribution of cases where each algorithm is used (see the colored bars where each color represents a different algorithm).

| Methods | CelebA | | | | MetaShift | | | |
|---|---|---|---|---|---|---|---|---|
| | ResNet18 | | CLIP (ViT-B/32) | | ResNet18 | | CLIP (ViT-B/32) | |
| | Alg. Selection | WG error ↓ | Alg. Selection | WG error ↓ | Alg. Selection | WG error ↓ | Alg. Selection | WG error ↓ |
| Oracle Selection | ▃▃▃▃▃ | 44.9 $_{\pm 0.2}$ | ▃▃▃▃▃ | 36.3 $_{\pm 0.3}$ | ▃▃▃▃▃ | 36.4 $_{\pm 0.3}$ | ▃▃▃▃▃ | 24.5 $_{\pm 0.3}$ |
| ERM | ▃ | 57.8 $_{\pm 0.4}$ | ▃ | 49.1 $_{\pm 0.2}$ | ▃ | 47.1 $_{\pm 0.3}$ | ▃ | 35.6 $_{\pm 0.3}$ |
| GroupDRO | ▃ | 52.5 $_{\pm 0.4}$ | ▃ | 45.7 $_{\pm 0.2}$ | ▃ | 45.0 $_{\pm 0.3}$ | ▃ | 33.4 $_{\pm 0.3}$ |
| Logits adjustment | ▃ | 53.1 $_{\pm 0.5}$ | ▃ | 41.4 $_{\pm 0.6}$ | ▃ | 40.5 $_{\pm 0.3}$ | ▃ | 28.0 $_{\pm 0.4}$ |
| Undersample | ▃ | 49.8 $_{\pm 0.5}$ | ▃ | 41.6 $_{\pm 0.2}$ | ▃ | 40.1 $_{\pm 0.3}$ | ▃ | 28.3 $_{\pm 0.3}$ |
| Oversample | ▃ | 52.4 $_{\pm 0.2}$ | ▃ | 45.8 $_{\pm 0.2}$ | ▃ | 45.1 $_{\pm 0.2}$ | ▃ | 33.2 $_{\pm 0.4}$ |
| OOD-Chameleon | ▃▃▃▃▃ | **47.7** $_{\pm 0.2}$ | ▃▃▃▃▃ | **39.1** $_{\pm 0.2}$ | ▃▃▃▃▃ | **39.0** $_{\pm 0.3}$ | ▃▃▃▃▃ | **27.2** $_{\pm 0.3}$ |

**Results.** In Table 2, we observe again that OOD-Chameleon performs well in selecting suitable algorithms for unseen tasks. Most importantly, the selector trained on a meta-dataset constructed from CelebA also performs well on MetaShift (Table 2), OfficeHome and Colored-MNIST (the latter two are in Appendix G) This means that **it learned generalizable relations between dataset properties and algorithms' performance, not idiosyncratic patterns specific to one dataset.**

In Table 3, we compare with single-algorithm baselines, i.e. using the same algorithm for all tasks. Our adaptive selection performs much better, confirming the premise that no single algorithm is a solution to all OOD scenarios. Interestingly, the proportions of cases where each algorithm is selected closely resemble those of the oracle (see the colored bars). In Appendix F, we evaluate our model with *estimated* dataset descriptors, e.g. for cases where attributes are not available at test time. The predictions of the selector remain accurate without further adaptation. In Appendix I, we show that a selector trained on datasets of smaller data sizes can generalize to larger ones; and we compare our approach with ensemble, which is much more expensive.

### 4.4. NLP Experiments

**Data.** The setup is analogous to the vision experiments (Section 4.3). We generate 720 tasks from

CivilComments (Borkan et al., 2019) and 180 from MultiNLI (Williams et al., 2017) (details in Appendix D).

**Results.** Table 4 and Table 18 in the appendix show that OOD-Chameleon is also effective in the language domain. The results closely align with those on vision tasks and confirm the findings from Section 4.3.

### 4.5. What Does OOD-Chameleon Learn?

Now we examine the decision rules learned by the selector. We first evaluate its complexity. To do so, we compare in Table 5 several implementations of the algorithm selector and make the following observations.

- First, a linear model performs significantly worse than an MLP. This shows that the MLP learns non-trivial rules and that there exist **non-linear relations** between dataset characteristics and the performance of algorithms.

- Second, a k-NN model is also significantly worse than an MLP. This shows that our model **works not only by memorizing a large number of example tasks**, which a k-NN also does. On the contrary, accurate predictions on unseen datasets require non-trivial generalization.

- Third, a simple decision tree performs almost as well as an MLP (details in Appendix J). Unlike an MLP, a tree is intrinsically interpretable, which offers the possibility of discovering new insights about existing algorithms.

*Table 4.* **Results on NLP tasks**. The selector is trained on a meta-dataset built from CivilComments, then evaluated on a separate set of tasks from CivilComments as well as MultiNLI. It remains accurate across datasets, meaning that it learns generalizable decision rules.

| Methods | CivilComments | | | | MultiNLI | | | |
| | BERT | | BERT (Finetuned) | | BERT | | BERT (Finetuned) | |
| | 0–1 ACC. ↑ | WG error ↓ | 0–1 ACC. ↑ | WG error ↓ | 0–1 ACC. ↑ | WG error ↓ | 0–1 ACC. ↑ | WG error ↓ |
|---|---|---|---|---|---|---|---|---|
| Oracle selection | 100 | 53.7 $_{\pm0.3}$ | 100 | 19.4 $_{\pm0.4}$ | 100 | 55.9 $_{\pm0.3}$ | 100 | 54.2 $_{\pm0.6}$ |
| Random selection | 14.5 $_{\pm0.7}$ | 66.6 $_{\pm0.4}$ | 50.0 $_{\pm0.4}$ | 23.5 $_{\pm0.2}$ | 19.8 $_{\pm0.3}$ | 67.1 $_{\pm0.2}$ | 20.5 $_{\pm0.5}$ | 63.6 $_{\pm0.7}$ |
| Global best | 29.9 $_{\pm0.8}$ | 62.7 $_{\pm0.4}$ | 55.9 $_{\pm1.2}$ | 22.9 $_{\pm0.3}$ | 34.4 $_{\pm0.4}$ | 63.6 $_{\pm0.6}$ | 27.2 $_{\pm0.6}$ | 61.4 $_{\pm0.5}$ |
| Naive descriptors | 68.8 $_{\pm1.9}$ | 57.2 $_{\pm0.4}$ | 81.1 $_{\pm0.8}$ | 21.7 $_{\pm0.3}$ | 64.3 $_{\pm0.8}$ | 59.2 $_{\pm0.4}$ | 68.2 $_{\pm0.5}$ | 57.4 $_{\pm0.4}$ |
| Regression | 51.3 $_{\pm1.0}$ | 57.6 $_{\pm0.4}$ | 47.9 $_{\pm0.7}$ | 22.6 $_{\pm0.4}$ | 57.7 $_{\pm0.8}$ | 59.4 $_{\pm0.2}$ | 40.8 $_{\pm0.3}$ | 60.6 $_{\pm0.2}$ |
| OOD-Chameleon | **81.9** $_{\pm1.1}$ | **55.8** $_{\pm0.4}$ | **90.9** $_{\pm1.4}$ | **20.7** $_{\pm0.2}$ | **79.4** $_{\pm0.6}$ | **58.3** $_{\pm0.2}$ | **74.4** $_{\pm0.9}$ | **56.6** $_{\pm0.2}$ |

*Table 5.* **Alternative implementations of the algorithm selector**. The lower performance of linear and k-NN models supports the importance of non-trivial, non-linear relations between dataset characteristics and algorithm performance (experiments on CelebA).

| Implementation | ResNet18 | | CLIP (ViT-B/32) | |
| | 0–1 ACC. ↑ | WG error ↓ | 0–1 ACC. ↑ | WG error ↓ |
|---|---|---|---|---|
| Linear | 63.4 $_{\pm1.0}$ | 49.6 $_{\pm0.3}$ | 67.6 $_{\pm1.8}$ | 42.1 $_{\pm0.3}$ |
| k-NN | 38.6 $_{\pm0.7}$ | 49.3 $_{\pm0.2}$ | 50.0 $_{\pm1.3}$ | 41.5 $_{\pm0.4}$ |
| Decision tree | 73.1 $_{\pm0.8}$ | 48.1 $_{\pm0.3}$ | 74.3 $_{\pm0.7}$ | 39.4 $_{\pm0.5}$ |
| **MLP** | **75.0** $_{\pm1.3}$ | **47.7** $_{\pm0.2}$ | **78.5** $_{\pm0.8}$ | **39.1** $_{\pm0.2}$ |

**Interpreting decision trees.** We visualize in Figures 10–11 the rules learned by selectors implemented as decision trees (see Figure 4 for a simplified illustration). Interpreting these trees leads to the following recommendations on the applicability of algorithms. (i) *Undersample* is preferred when the size of the dataset is not too small and the distribution shifts are severe. (ii) When there are indeed only a small number of samples but relatively large shifts, one could resort to *Logits adjustment*. This aligns with Nguyen et al. (2021) who suggested that *Logits adjustment* should be more effective than *Undersample* when the number of unique samples is low for minority groups. (iii) *GroupDRO* is more useful when there are enough samples and relatively large distribution shifts. (iv) For mild shifts, *ERM* or *Oversample* are the best options. Overall, such observations provide rich material for future investigations on the applicability of existing learning algorithms.

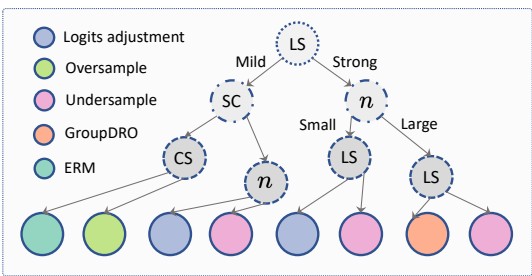

*Figure 4.* **Simplified illustration of a learned model selector implemented as a decision tree** (see Figures 10–11 for full versions).

**Attributing algorithm effectiveness to data characteristics.** To complement the general rules revealed by decision trees, we look at factors affecting local decisions **between any two algorithms**. We perform a leave-one-descriptor-out training of *pairwise* algorithm selectors, i.e. which consider only two candidate algorithms. We plot the results in Figure 5. For a chosen pair of algorithms, each bar shows a drop in performance relative to the full descriptors (leftmost bar). This drop indicates the importance of a specific piece of information for distinguishing the two algorithms. In the case of *Oversample/Undersample* for example, the data size ($n$) and degree of spurious correlation ($d_{sc}$) are the most important. This observation is consistent with the analysis in (Nguyen et al., 2021) that implies that, while undersampling can cope with more distribution shifts than oversampling, it is inferior when the number of samples in the minority group is small (i.e. when $n$ or $d_{sc}$ is too small).

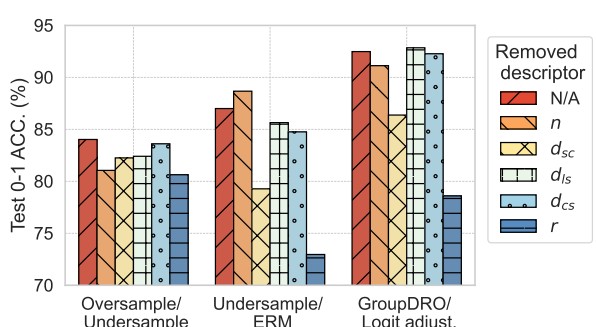

*Figure 5.* **Leave-one-descriptor-out training of pairwise algorithm selectors.** The drops in accuracy reveal the importance of various factors in the suitability of one algorithm vs. another.

We also observe that the importance of the different pieces of information varies for other pairs of algorithms (Figure 5 middle/right bars). We can thus identify which characteristics of the data are important *in practice* for specific algorithms to outperform one another. *These are rich observations for future work that could confront them with existing theories on the applicability of these algorithms.*

# 5. Related Work

**OOD generalization** is a wide field of study, see e.g. Liu et al. (2021b) for a survey. The conclusion from several studies is that there is no one-fits-all solution to distribution shifts (Gulrajani & Lopez-Paz, 2020; Wiles et al., 2021; Nguyen et al., 2021; Ye et al., 2022; Liang & Zou, 2022; Yang et al., 2023; Bell et al., 2024). Indeed, a standard ERM baseline is generally very effective, but there also exist a variety of training algorithms with demonstrated benefits in *specific* OOD settings. This motivates our goal of improving the selection among existing algorithms, which we seek to automate with a data-driven approach.

**Algorithm selection.** Model selection (Forster, 2000; Raschka, 2020) and algorithm selection (Rice, 1976; Kerschke et al., 2019) are integral parts of machine learning workflows. In the context of OOD generalization, recent work (Liu et al., 2023b; Baek et al., 2022; Garg et al., 2022; Miller et al., 2021; Lu et al., 2023) proposes heuristics to predict a models' OOD performance on unlabeled target data. These heuristics have downsides: (i) they can only be estimated after training one or several models, (ii) they require (unlabeled) test data, and (iii) they rely on unverifiable assumptions (Teney et al., 2023). A major difference of our work is to aim for *a priori* selection of the best algorithm, i.e. before training on the target data.

Concurrent work by Bell et al. (2024) identifies a motivation similar to ours. They propose to select among algorithms to deal specifically with spurious correlations, based on the algorithms' past performance on benchmarks similar to the target data. The differences with our work are: (i) we use a *learning* approach rather than a simple similarity; (ii) our learning process relies on *semi-synthetic* data rather than existing benchmarks, which helps cover a broader set of distribution shifts; (iii) we consider *label shifts*, *covariate shifts* and their combinations, not only spurious correlations.

**Meta-learning.** Our approach learns from a "dataset of datasets" i.e. a distribution of tasks, which is a principle of meta-learning (Vilalta & Drissi, 2002; Hospedales et al., 2022). Most similarly to us, Achille et al. (2019); Öztürk et al. (2022); Arango et al. (2023); Zhang et al. (2023) learn to select among pretrained models for downstream tasks, or for outlier detection Zhao et al. (2021). Contrary to us, these works first require training multiple candidate models and they do not target OOD generalization.

**AutoML.** Our approach relates to AutoML, which aims to automate ML workflows, often by trial-and-error (Hutter et al., 2019; He et al., 2021). Our work differs in that (1) it aims for *a priori* algorithm selection; (2) it can provide new insights on the applicability of existing algorithms.

# 6. Conclusions

This paper highlighted the importance of algorithm selection for OOD generalization. We described a data-driven solution to automate this selection, by turning it into a standard supervised multi-label classification problem. Crucially, we demonstrated empirically that training our algorithm selector on a collection of semi-synthetic datasets is sufficient to learn decision rules that generalize to unseen shifts and datasets. This is a strong empirical result, since the model could as well have overfitted to idiosyncratic properties of example shifts and datasets. Instead, it learns generalizable relations between dataset properties and algorithms' performance. We verified that the selection relies on non-trivial patterns, and also ruled out the possibility that it simply memorizes a large number of example cases.

**Limitations and future work.**

- **Algorithm baselines**: We focused on algorithms known to be reliable, simple but effective in different settings (and often superior to fancier ones, see Idrissi et al. (2022)). Future extension will include more algorithms and their variations, e.g. by including some of their hyperparameters in the search space.

- **Learnable dataset descriptors**: We hand-designed interpretable dataset descriptors, but they depend on attribute annotations and might miss important information. The next step is to learn descriptors end-to-end with the selector, e.g. with Set Transformers or Dataset2Vec (Lee et al., 2019; Jomaa et al., 2021).

- **Generalization to other shift types**: We considered three types of shifts (spurious correlations, label shifts, covariate shifts) and their combination, which covers most, if not all, shifts studied in the literature. However, generalizing to other shift types might require additional curation. For example, covariate shifts can happen in many forms except for the attribute distribution shifts we studied.

- **Generalization to broader scenarios**: Our tool to quantify distribution shifts applies to binary classification with binary attributes, which is the default setting in studying spurious correlations. Extension to multiple classes and multiple attributes is valuable. Besides, optimizing a combination of multiple relevant performance metrics, instead of a single one, is promising.

- **Real-world evaluation**: A crucial next step is to study how the algorithm selector trained on a semi-synthetic dataset collection generalizes to real-world tasks, such as WILDS (Koh et al., 2021; Sagawa et al., 2022).

- **Understanding algorithms' applicability**: We extracted insights from the trained selector, which could be used to form the basis of future investigations on the applicability and inductive bias of many other algorithms.

## Acknowledgements

We thank Caglar Gulcehre, Devis Tuia and Thiên-Anh Nguyen for the helpful discussions, and the anonymous reviewers for their thoughtful feedback.

## Impact Statement

This paper presents work whose goal is to advance the field of Machine Learning. There are many potential societal consequences of our work, none of which we feel must be specifically highlighted here.

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

# Appendix

The appendix provides the following additional details and results.

- Appendix A provides an **overview of the used source datasets**.
- Appendix B details the **construction of datasets with desired distribution shifts**.
- Appendix C describes the **data and detailed setup of the controlled experiments** of Section 4.2.
- Appendix D describes the **data and detailed setup of the realistic experiments** of Sections 4.3 and 4.4.
- Appendix E contains various **ablation studies** (hyperparameters, training paradigms, etc).
- Appendix F presents **additional results on the use of estimated dataset descriptors** as input to the selector.
- Appendix G presents **additional results on vision tasks** of Section 4.3.
- Appendix H presents **additional results on language tasks** of Section 4.4.
- Appendix I presents **additional results on efficiency considerations**.
- Appendix J shows **additional details of the algorithm selector implemented by a decision tree** of Section 4.5.

## A. Dataset Overview

In our experiments, we create tasks with distribution shifts by subsampling from 6 real-world datasets from computer vision and natural language processing domains. Here we provide details for each of them. Some content in this section is borrowed from (Yang et al., 2023), and we appreciate their clean and concise dataset summary. For visual examples, see Table 6 and Table 7.

**CelebA.** CelebA (Liu et al., 2015) is a large-scale face attributes dataset commonly used in facial recognition, attribute prediction, and generative modeling research. It contains over 200,000 celebrity images covering approximately 10,000 identities, each annotated with 40 binary facial attributes (e.g., smiling, eyeglasses, blond hair) and 5 landmark locations (e.g., eyes, nose, mouth corners). The images exhibit substantial variability in pose, lighting, background, and occlusion. The standard task involves predicting facial attributes or generating/editing facial images using these annotations. The dataset is publicly available for non-commercial research use.

**MetaShift.** MetaShift (Liang & Zou, 2022) is a general method of creating image datasets from the Visual Genome project. Here, we make use of the pre-processed Cat *vs.* Dog dataset, where the goal is to distinguish between the two animals. The spurious attribute is the image background, where cats and more likely to be indoors, and dogs are more likely to be outdoors. We use the "unmixed" version generated from the authors' codebase.

**OfficeHome.** OfficeHome (Venkateswara et al., 2017) is a multi-class image classification dataset commonly used in domain adaptation research. It consists of around 15,500 images across 65 object categories (e.g., keyboard, pen, mug), drawn from four visually distinct domains: *Art*, *Clipart*, *Product*, and *Real-World*. The standard task is to classify the object category across varying domain pairs. The dataset is publicly available for non-commercial research use.

**Colored-MNIST.** Colored-MNIST (Arjovsky et al., 2020) is a synthetic variant of the MNIST dataset commonly used to study spurious correlations and domain generalization. It is constructed by coloring the grayscale MNIST digits with label-dependent or label-independent color schemes to introduce a spurious correlation between digit class and color. Multiple environments are created by varying the strength of this correlation, allowing researchers to evaluate models' robustness to distribution shifts. The standard task is digit classification under different color-based domain shifts.

**CivilComments.** CivilComments (Borkan et al., 2019) is a binary classification text dataset, where the goal is to predict whether a internet comment contains toxic language. The spurious attribute is whether the text contains reference to eight demographic identities (*male, female, LGBTQ, Christian, Muslim, other religions, Black,* and *White*).

**MultiNLI.** MultiNLI (Williams et al., 2017) is a text classification dataset with 3 classes, where the target is the natural language inference relationship between the premise and the hypothesis (neutral, contradiction, or entailment). The spurious attribute is whether negation appears in the text, as negation is highly correlated with the contradiction label.

*Table 6.* Example inputs for Colored-MNIST (Arjovsky et al., 2020), CelebA (Liu et al., 2015), MetaShift (Liang & Zou, 2022) and OfficeHome (Venkateswara et al., 2017). The table is adapted from (Yang et al., 2023).

| Dataset | Examples | | | | | |
|---------|----------|--|--|--|--|--|
| Colored-MNIST |  | | | | | |
| CelebA |  | | | | | |
| MetaShift |  | | | | | |
| OfficeHome |  | | | | | |

*Table 7.* Example inputs for CivilComments (Borkan et al., 2019) and MultiNLI (Williams et al., 2017). The table is borrowed from (Yang et al., 2023).

| Dataset | Examples |
|---------|----------|
| CivilComments | "Munchins looks like a munchins. The man who dont want to show his taxes, will tell you everything..." 
 "The democratic party removed the filibuster to steamroll its agenda. Suck it up boys and girls." 
 "so you dont use oil? no gasoline? no plastic? man you ignorant losers are pathetic." |
| MultiNLI | "The analysis proves that there is no link between PM and bronchitis." 
 "Postal Service were to reduce delivery frequency." 
 "The famous tenements (or lands) began to be built." |

# B. Construction of the Meta-dataset

In Section 3, we describe a framework that allows for constructing datasets with diverse distribution shifts by sampling from synthetic distributions or existing datasets, here we provide more details and examples. There are two use cases in Section 4.2, Section 4.3&4.4 respectively, where in both cases we know the distribution of each group (recall that the combinations of different values of attribute $a$ and class $y$ form different groups). Specifically, in Section 4.2, we have the group distributions as Gaussian distributions so that we can sample the desired numbers of samples from those distributions, while in Section 4.3 and 4.4 we have a decent amount of samples in each group of the fine-grained annotated real-world datasets used (CelebA, MetaShifts, OfficeHome, Colored-MNIST for vision tasks and CivilComments, MultiNLI for NLP tasks), and we can then also sample the desired numbers of samples from each group.

In Section 3, we define the degrees of distribution shifts as a function of the number of samples for each group. Therefore, to obtain a dataset with specific degrees of distribution shifts, **one only needs to solve the number of samples for each group and sample them from the group distribution**. Note that the number of samples can be scaled up or down depending on the expected size of the dataset. All of the constraints to be solved are therefore:

$$d_{sc} = \frac{|\mathcal{G}_1| + |\mathcal{G}_4|}{\sum_i |\mathcal{G}_i|}, \quad d_{ls} = \frac{|\mathcal{G}_1| + |\mathcal{G}_3|}{\sum_i |\mathcal{G}_i|}, \quad d_{cs} = \frac{|\mathcal{G}_1| + |\mathcal{G}_2|}{\sum_i |\mathcal{G}_i|},$$
$$\sum_i |\mathcal{G}_i| = n, \quad (|\mathcal{G}_i| \geq 0) \tag{4}$$
$$0 \leq d_{sc} \leq 1, \quad 0 \leq d_{ls} \leq 1, \quad 0 \leq d_{cs} \leq 1,$$

Solving the constraints gives the feasible solution set of the degrees of distribution shifts, as shown in Figure 6. We see that not any value in the cube can be chosen because of the constraint $|\mathcal{G}_i| \geq 0$ for $\forall i$.

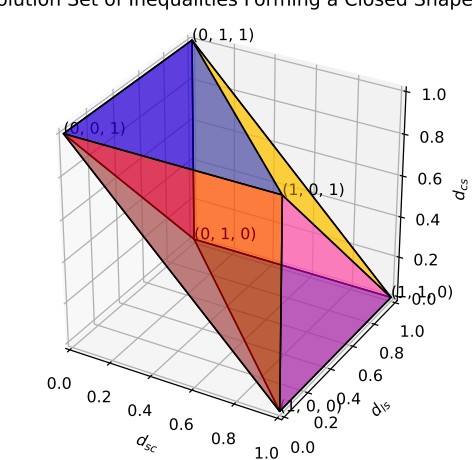

*Figure 6.* The feasible degrees of spurious correlation ($d_{\mathrm{sc}}$), covariate shift ($d_{\mathrm{cs}}$) and label shift ($d_{\mathrm{ls}}$).

**Generating datasets with diverse characteristics.**  We then generate datasets with a desired size, distribution shifts, and availability of the spurious feature. More precisely, we vary the following.

- The degrees of distribution shifts (as long as they are picked from the feasible solution set in Figure 6).
- The size of the dataset.
- The group distributions (for simulating different availabilities of the spurious feature). For synthetic experiments, they are more controllable in that the group distributions are known and the availability of spurious feature is well-defined (see Section 4.2). For vision tasks and NLP tasks, we respectively use CelebA and CivilComments to create the dataset of datasets. Since they are annotated with various labels, we can use different pairs of them as label $Y$ and spurious attribute $A$. See Table 9 on the pairs we choose. Each different label-attribute pair results in different group distributions, and how "spurious" the spurious attribute is.
- For the synthetic experiments (Section 4.2), we additionally vary the dimensionality of the data points $d$ (See Table 8).

See Figure 7 for example tasks generated from CelebA. We summarize the characteristics of the generated datasets in the following sections.

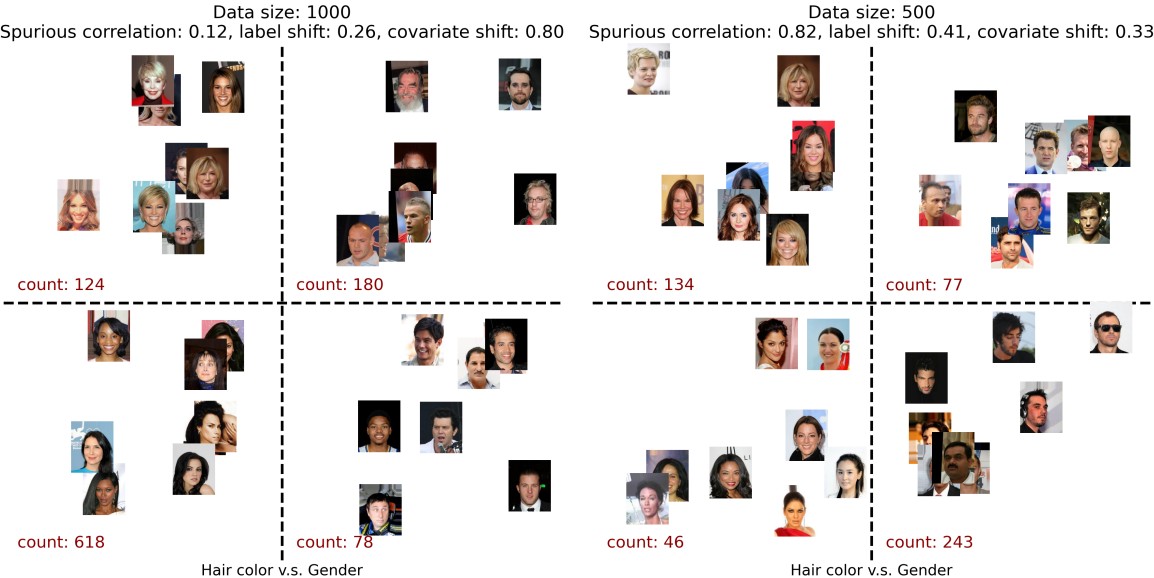

*Figure 7.* Example datasets (only training set is shown for each) with different data size and distribution shifts, generated from CelebA. The meta-dataset is composed of many more such datasets, each exhibiting different characteristics.

## C. Controlled Experiments: Datasets and Detailed Setup

We provide details on the experimental setup of the controllable experiments of Section 4.2.

**Dataset of datasets.**  In total, we created 9,240 tasks, see Table 8 for the statistics for these tasks. Specifically, we generate the training sets of the tasks with the combinations of values in Table 8, and for the degrees of distribution shifts, we consider 2 cases: (1) 3 shifts (spurious correlation, covariate shift and label shift) all present each in different degrees, we uniformly sampled 30 different triple degrees, and (2) there is only 1 shift present, in this case for each shift we consider 9 different values shown in the table (therefore in total $3 * 9 - 2 = 25$ different triple degrees). We generate their test sets with the same number of samples as its training set for each dataset, but keep the number of samples the same for all groups (therefore balanced test sets with all $d_{(.)} = 0.5$). We randomly split these datasets into 4:1 as the dataset of datasets for training the algorithm selector, and the unseen datasets for evaluation.

*Table 8.* Statistics for the dataset of datasets in controllable synthetic experiments.

| Statistics | Value |
|---|---|
| Size of training set | $\{200, 500, 1000, 2000, 3000, 5000, 10000\}$ |
| Input dimensionality | $\{2, 10, 50, 100\}$ |
| Availability $r = \sigma_c^2/\sigma_a^2$ | $\{1, 5, 10, 20, 50, 100\}$ |
| 3 shifts | randomly sampled 30 feasible degrees (Figure 6) |
| 1 shift | $\{0.01, 0.05, 0.1, 0.3, 0.5, 0.7, 0.9, 0.95, 0.99\}$ |

**Algorithm selector.**  We use 4-layer MLPs to parameterize the algorithm selector, because we found a shallower or simpler model underfits while a deeper model does not provide more improvements.

**Tasks.**  To solve the tasks defined by each dataset in the dataset of datasets, we use Adam optimizer with default hyperparameters, along with L2 regularization. We train for 1000 epochs to ensure convergence on this synthetic example.

## D. Datasets and Detailed Setup of the Vision and NLP Experiments

### D.1. Availability of the Spurious Features

In the controllable experiments in Section 4.2, we compute the availability of spurious features as $r = \sigma_c^2/\sigma_a^2$, following the spurious-core information ratio defined in Sagawa et al. (2020). The higher the $r$, the more signal there is about the spurious attribute in the spurious features, relative to the signal about the label in the core features.

However, in real-world data (Section 4), it is *not* straightforward how to generate the dataset of datasets with diverse $r$. In Section 4.3, we use different attribute pairs in CelebA (Liu et al., 2015) (one as the label attribute, i.e. the attribute we want to classify, and the other as spurious attribute) as shown in Table 9 (Left), to account for different availability of spurious features. Intuitively, the obviousness of different types of attributes (in terms of size, color, etc) signifies different availability of the spurious feature (caused by the attribute).

Similarly, we use different attribute pairs in CivilComments (Borkan et al., 2019), as shown in Table 9 (Right) to ensure a diverse availabilities of spurious feature in the created tasks.

*Table 9.* Attribute pairs in CelebA (Liu et al., 2015) (Left) and CivilComments (Borkan et al., 2019) (Right) that are used to construct different availability of the spurious feature.

| CelebA | | CivilComments | |
|---|---|---|---|
| Label | Spurious attribute | Label | Spurious attribute |
| Mouth Slightly Open | Wearing Lipstick | | Male |
| Attractive | Smiling | Toxicity | Female |
| Black Hair | Male | | Black |
| Oval Face | High Cheekbones | | White |

By using different attribute pairs, we are able to generate tasks with different availabilities. However, unlike the controllable experiments, here there is no straightforward way of quantifying the availability of spurious feature. Therefore, as mentioned in Section 4.3, we use a proxy availability which is the average distances of the samples' embeddings to their labels or attributes clustering center, respectively. Specifically, we first obtain the samples embeddings $\{z_i\}_{i=1}^{n_{tr}}$ from the used backbone (ResNet18/CLIP or BERT/BERT (fine-tuned)) and then compute the clustering center of each label and attribute as $\boldsymbol{\mu}_y$ and $\boldsymbol{\mu}_a$. The availability $r$ is then:

$$
\begin{aligned}
r &= \frac{\sum_y l_y}{\sum_a l_a} \\
l_y &= ||z_i - \boldsymbol{\mu}_y||_2 \\
l_a &= ||z_i - \boldsymbol{\mu}_a||_2 \\
\boldsymbol{\mu}_y &= \frac{1}{|y_i = y|} \sum_{y_i = y} z_i \\
\boldsymbol{\mu}_a &= \frac{1}{|a_i = a|} \sum_{a_i = a} z_i
\end{aligned}
\tag{5}
$$

This definition is intuitively similar to what is defined for availability in controllable experiments, where we had $r = \sigma_c^2/\sigma_a^2$. Intuitively, a smaller average distance w.r.t. attributes signify an easier spurious feature and therefore, higher availability. Only with Equation 5, we are able to compute availability for any tasks, and thus enables transferring the algorithm selector trained from CelebA tasks, to unseen tasks from other datasets (i.e. MetaShift (Liang & Zou, 2022), OfficeHome (Venkateswara et al., 2017), Colored-MNIST (Arjovsky et al., 2020) and MultiNLI (Williams et al., 2017))

### D.2. Detailed Experimental Setup

We provide the detailed experimental setup for the vision experiments in Section 4.3 and NLP experiments in Section 4.4.

**Dataset of datasets.** In total, we create 720 tasks from CelebA (Liu et al., 2015), see Table 10 for the statistics for these datasets. Specifically, we generate the training sets of the datasets with the combinations of values in Table 10, and for the degrees of distribution shifts, we consider 2 cases: (i) 3 shifts (spurious correlation, covariate shift and label shift) all present each in different degrees, we uniformly sample 30 different triple degrees, and (ii) there is only 1 shift present, in this case for each shift we consider 10 different values (therefore in total 30 different triple degrees). We generate their test sets with half the number of samples as their training set for each dataset, but keep the number of samples the same for all groups (therefore balanced test sets with all $d_{(.)} = 0.5$). We randomly split the generated CelebA tasks into 4:1 training and test splits, and use the training part for the meta-dataset $\mathbb{D}$.

For evaluation, we first evaluate on the unseen tasks from CelebA. In addition, we create 129 datasets from MetaShift (Liang & Zou, 2022) ({cat, dog} and {indoor, outdoor} split) and 180 datasets from Colored-MNIST (Arjovsky et al., 2020) for evaluation, where we follow the same generation procedure. The generated number of tasks for MetaShift and Colored-MNIST is smaller because: (i) These two datasets only have one attribute annotated (indoor/outdoor and colors, respectively), so we can only consider 1 attribute-label pair and hence one availability of spurious features. (ii) Their number of samples per group for MetaShift is not as sufficient as CelebA, making some of the datasets infeasible (i.e. the Equation 4 as no solution for some dataset characteristics). We also create 100 tasks from OfficeHome for evaluation, from a slightly different procedure, see OfficeHome paragraph in Appendix G.

For NLP experiments, we follow a similar process and generated 720 tasks from CivilComments (Borkan et al., 2019) and generat 180 tasks from MultiNLI (Williams et al., 2017).

**Algorithm selector.** We use a 2-layer MLP (with hidden size 100) to parameterize the algorithm selector, because we found a shallower or simpler model underfits while a deeper model does not provide more improvements.

**Tasks.** To solve the tasks defined by each dataset in the dataset of datasets, we either do (1) linear probing on ResNet18 or CLIP (ViT-B/32) or (2) fine-tuning ResNet18. In the first case (all tables except for Table 11), we use Adam optimizer with default hyperparameters, along with L2 regularization. We train for 1000 epochs to ensure convergence. In the second case (Table 11), we use SubpopBench (Yang et al., 2023) and its default hyperparameters to fine-tune ResNet18. We use basic data augmentations (resize, crop, etc.).

*Table 10.* Statistics for the dataset of datasets in vision and NLP experiments.

| Statistics | Value |
|---|---|
| Size of training set | $\{200, 500, 1000\}$ |
| Input dimensionality | N/A |
| Availability | 4 different attribute pairs, see Table 9 |
| 3 shifts | randomly sampled 30 feasible degrees (Figure 6) |
| 1 shift | randomly sampled 10 degrees in $(0, 1)$ for each type of shifts |

*Table 11.* **Evaluation of training paradigms for tasks**. The algorithm selector generalizes well with both linear probing and fine-tuning.

| Methods | Linear probing | | Fine-tuning | |
|---|---|---|---|---|
| | 0–1 ACC. ↑ | WG error ↓ | 0–1 ACC. ↑ | WG error ↓ |
| Oracle Selection | 100 | 44.9 $_{\pm 0.2}$ | 100 | 32.3 $_{\pm 0.4}$ |
| Regression | 53.6 $_{\pm 1.1}$ | 49.8 $_{\pm 0.5}$ | **73.6** $_{\pm 1.8}$ | **34.5** $_{\pm 0.3}$ |
| OOD-Chameleon | **75.0** $_{\pm 1.3}$ | **47.7** $_{\pm 0.2}$ | 67.8 $_{\pm 1.3}$ | 34.8 $_{\pm 0.4}$ |

## E. Ablation Studies

### E.1. Different Training Paradigms

Here we conduct an ablation study of the training paradigms for the tasks. This is necessary to check because, solving the tasks by different training paradigms affects the algorithms' OOD performance $P_{jm}$ and therefore changes the distribution of meta-dataset $\mathbb{D} = \{f(D_j^{\mathrm{tr}}), \mathcal{A}_m, P_{jm}\}$. In Table 11, we verify that the algorithm selector works well with both linear probing and fine-tuning with CelebA tasks. The results indicate that the learned algorithm selector can also accurately select suitable algorithms when the models for the tasks become over-parametrized in the case of fine-tuning.

### E.2. Algorithm Suitability Threshold $\epsilon$

In Section 2.2, we convert the algorithm selection problem into a multi-label classification task, because multiple algorithms can be suitable for a given dataset/task. This involves defining a small threshold $\epsilon$. Specifically, an algorithm is considered suitable if $(P_{jm} - \min_m P_{jm}) \leq \epsilon$ for a small threshold $\epsilon$. We have used $0.05$ for all our experiments, and here we provide an ablation study on it, in Table 12. The method proves robust to other choices than the $\epsilon = 0.05$ used throughout the paper. Intuitively, when $\epsilon$ is too big, the meta-dataset loses discriminative power; too small, it is subject to noise.

*Table 12.* Ablation study on $\epsilon$. Worst-group error is shown (lower is better).

| $\epsilon$ | CelebA (ResNet) | CelebA (CLIP) | etaShift (ResNet) | MetaShift (CLIP) |
|---|---|---|---|---|
| 0.0 | 48.3 | 39.8 | 39.9 | 27.7 |
| 0.025 | 48.1 | 39.6 | 39.7 | 27.3 |
| 0.05 | **47.7** | **39.1** | **39.0** | **27.2** |
| 0.10 | 48.0 | 39.4 | 39.6 | 27.8 |

### E.3. Algorithm Selection Strategies at Test Time

As discussed in Section 2.2, multiple algorithms can be predicted as suitable for a dataset/task at test time, due to our multi-label classification formulation. When this is indeed the case, we always select the algorithm with the highest prediction logits, which corresponds to the one the algorithm selector is the most confident in. An alternative would be to use binary predictions, and if multiple algorithms are predicted to be suitable, randomly select one of them. Here we show that, empirically, the former option (top logits) is much better, in Table 13.

*Table 13.* **Comparison of test-time algorithm selection criteria**. Worst-group error is shown, lower is better.

| Selection criterion | CelebA (ResNet) | CelebA (CLIP) | MetaShift (ResNet) | MetaShift (CLIP) |
|---|---|---|---|---|
| Top logits (used elsewhere) | 47.7 | 39.1 | 39.0 | 27.2 |
| Binary predictions | 48.5 | 39.8 | 39.4 | 27.4 |

### E.4. Performance Metrics

In our experiments, we mostly use the worst-group (WG) error as the performance metric. The rationale is that it is independent of the test distribution and can thus be addressed by considering only the distribution (imbalances) of training data. WG performance thus promotes general robustness to distribution shifts and has been used in many prior works.

However, our formulation is compatible with other performance metrics. The algorithm selection should work as long as there are useful relations to be learned between the data descriptors, the algorithms and the performance metrics. Here we investigate an alternative metric, the averaged-group error, in Table 14. The conclusions are similar to those in Section 4 – the algorithm selector can still select suitable algorithms with averaged-group error as the performance metric.

*Table 14.* **Training and evaluating the algorithm selector with averaged-group error**. Lower is better.

| | CelebA (ResNet) | CelebA (CLIP) | MetaShift (ResNet) | MetaShift (CLIP) |
|---|---|---|---|---|
| Oracle selection | 33.8 | 26.4 | 24.6 | 14.8 |
| Random selection | 36.6 | 29.5 | 27.8 | 18.1 |
| Global best | 35.7 | 28.0 | 26.9 | 16.6 |
| Naive descriptors | 35.8 | 27.7 | 26.7 | 16.9 |
| Regression | 35.5 | 28.3 | 27.0 | 16.4 |
| OOD-Chameleon | **35.1** | **27.2** | **26.3** | **15.9** |

## F. Algorithm Selection with Estimated Dataset Descriptors

Here we study the scenarios where the information to compute dataset descriptors, such as the attribute for each sample (Liu et al., 2021a), cannot be obtained at test time. Recent works (Liu et al., 2021a; Kirichenko et al., 2022; Qiu et al., 2023; Lee et al., 2022; Pagliardini et al., 2022) for OOD generalization aim to eliminate the need for attribute annotation by either: (i) infer the attribute annotation and then use them to run algorithms that require attribute annotation, (ii) run ERM on the training set assuming that the ERM learns the spurious feature, and then build invariant classifier on top of the ERM classifier (e.g. fit a model that disagrees with the ERM).

We leverage the above first line of research, i.e. inferring the attribute annotation and use them to compute the dataset descriptors on the target training set. Then, we can use OOD-Chameleon to select the suitable algorithms. We infer the attributes by clustering the embeddings from frozen backbones, following Sohoni et al. (2020); You et al. (2024). The intuition is that different attributes, such as cows in grass or desert, can be considered as 'subclasses' or 'hidden stratifications', and they are observed to be separable in the feature space of the deep models. Hence, for example we can infer which cows belong to which environment, by clustering on their embeddings. In particular, the training samples are passed through the backbone of the task (i.e. ResNet18 or CLIP in our case), and get the embeddings. For each semantic class, we cluster the corresponding samples with K-means and assign different attributes to different clusters. This gives each sample its inferred attribute annotation. We can then use these inferred attribute annotations to compute dataset descriptors, where they are used to compute the degree of spurious correlation $d_{sc}$ and covariate shift $d_{cs}$, as well as the availability of spurious features $r$ (see Appendix D.1 on how we compute the availability).

With the inferred dataset descriptors, we use the algorithm selector to predict suitable algorithms. In Tab 15, we show that the suitable algorithms are still predictable with estimated dataset descriptors. Interestingly, while having performance drops in most cases, using estimated dataset descriptors boosts the performance on MetaShift with ResNet18.

*Table 15.* OOD-Chameleon with inferred attributes (and dataset descriptors) on CelebA and MetaShift. 0–1 Accuracy (higher is better) is shown.

| Methods | Attribute | CelebA | | MetaShift | |
|---|---|---|---|---|---|
| | | ResNet18 | CLIP (ViT-B/32) | ResNet18 | CLIP (ViT-B/32) |
| Oracle Selection | N/A | 100 | 100 | 100 | 100 |
| Regression | ✓ | 53.6 $_{\pm1.1}$ | 46.7 $_{\pm0.9}$ | 51.2 $_{\pm0.3}$ | 49.2 $_{\pm0.3}$ |
| OOD-Chameleon | ✓ | 75.0 $_{\pm1.3}$ | 78.5 $_{\pm0.8}$ | 80.6 $_{\pm0.7}$ | 76.7 $_{\pm0.5}$ |
| Regression | ✗ | 52.1 $_{\pm0.9}$ | 50.2 $_{\pm1.2}$ | 53.7 $_{\pm0.6}$ | 47.7 $_{\pm1.0}$ |
| OOD-Chameleon | ✗ | 73.2 $_{\pm0.6}$ | 72.0 $_{\pm1.1}$ | 83.4 $_{\pm0.8}$ | 72.6 $_{\pm0.5}$ |

# G. Additional Results on Vision Tasks

**Colored-MNIST.** In Section 4.3, we see that when training the algorithm selector on tasks built from CelebA, the algorithm selector is not only able to select suitable algorithms on unseen tasks of CelebA but also on unseen tasks of MetaShift. This suggests the learned data/algorithm relation is robust and transferrable across data domains. Here we provide more experiments on Colored-MNIST as a further support, in particular, we train on the same CelebA meta-dataset and evaluate the algorithm selector on Colored-MNIST tasks (Arjovsky et al., 2020). In Colored-MNIST dataset, there are images of 10 digits from 0-9 and the digits are divided into 2 classes (i.e. 0–4 is class 0, 5–9 is class 1). In addition, the two classes of digits are in two different colors (e.g. red and green). When the colors of the digits correlate with the shapes of digits, a spurious correlation occurs. We create 180 tasks from Colored-MNIST, each task exhibits different magnitudes of spurious correlation (SC), label shift (LS) and covaraite shifts (CS) and the size of datasets span {200, 500, 1000}. In Table 16, we see that the algorithm selector proves to be effective on Colored-MNIST as well.

*Table 16.* **Results on algorithm selection for unseen Colored-MNIST tasks**. Algorithm selectors are trained on the meta-dataset generated from CelebA and evaluated on unseen Colored-MNIST tasks. ResNet18 and CLIP (ViT-B/32) refer to the models used in the tasks.

| Methods | Colored-MNIST | | | |
|---|---|---|---|---|
| | ResNet18 | | CLIP (ViT-B/32) | |
| | 0–1 ACC. ↑ | WG error ↓ | 0–1 ACC. ↑ | WG error ↓ |
| Oracle Selection | 100 | 21.1 $_{\pm0.3}$ | 100 | 13.3 $_{\pm0.4}$ |
| Random Selection | 29.3 $_{\pm1.1}$ | 28.1 $_{\pm0.3}$ | 39.3 $_{\pm0.5}$ | 19.0 $_{\pm0.3}$ |
| Global Best | 50.1 $_{\pm0.7}$ | 25.1 $_{\pm0.4}$ | 63.3 $_{\pm1.2}$ | 16.0 $_{\pm0.2}$ |
| Regression | 79.4 $_{\pm0.5}$ | 23.8 $_{\pm0.3}$ | 54.2 $_{\pm0.9}$ | 16.1 $_{\pm0.4}$ |
| OOD-Chameleon | **82.7** $_{\pm0.8}$ | **23.5** $_{\pm0.4}$ | **75.3** $_{\pm0.7}$ | **15.6** $_{\pm0.4}$ |

**OfficeHome.** We additionally verify our approach with OfficeHome (Venkateswara et al., 2017). In OfficeHome, the there exists 65 classes of objects (such as Mug, Pen, Spoon, Knife, and other objects typically found in Office and Home settings.) and each of the classes have samples in 4 different domains: Artistic images, Clip Art, Product images and Real-World images. The samples of each domain and object pair are uneven.

We sampled tasks from OffceHome by randomly sampling pairs of domain and object (e.g. {pen, knife} in {Art, Clipart}), and then collect the corresponding samples from OfficeHome. We created 100 tasks in total, and evaluate the trained algorithm selector from CelebA on these tasks, in Table 17. The results show that OOD-Chameleon is still able to select suitable algorithms, evidenced by the lowest worst-group error across the 100 tasks. This provides strong additional support for the utility of our approach to select suitable algorithms across datasets and diverse types of shifts.

*Table 17.* **Results on algorithm selection for unseen OfficeHome tasks** (worst-group error is shown, lower is better). Algorithm selectors are trained on the meta-dataset generated from CelebA and evaluated on unseen OfficeHome tasks. ResNet18 and CLIP (ViT-B/32) refer to the models used in the tasks.

|  | Office-Home (ResNet) | Office-Home (CLIP) |
|---|---|---|
| Oracle selection | (14.8) | (11.4) |
| Random selection | 19.3 | 15.1 |
| Global best | 18.2 | 14.4 |
| Naive descriptors | 18.8 | 14.6 |
| Regression | 18.5 | 14.3 |
| OOD-Chameleon | **17.9** | **13.5** |

# H. Additional Results on NLP Tasks

In Figure 18, we show a comparison with single-algorithm baselines, i.e. using the same algorithm for all tasks, on CivilComments and MultiNLI tasks. Similar to vision experiments (Table 3), our adaptive selection performs much better, confirming the premise that no single algorithm is a solution to all OOD scenarios.

*Table 18.* **Comparison with static algorithm selection.** Observations are similar to Table 3.

| Methods | CivilComments | | | | MutliNLI | | | |
|---|---|---|---|---|---|---|---|---|
| | BERT | | BERT (Finetuned) | | BERT | | BERT (Finetuned) | |
| | Alg. Selection | WG error ↓ | Alg. Selection | WG error ↓ | Alg. Selection | WG error ↓ | Alg. Selection | WG error ↓ |
| Oracle Selection | | $53.7_{\pm 0.3}$ | | $19.4_{\pm 0.4}$ | | $55.9_{\pm 0.3}$ | | $54.2_{\pm 0.6}$ |
| ERM | | $78.2_{\pm 0.4}$ | | $28.2_{\pm 0.7}$ | | $76.1_{\pm 0.6}$ | | $68.2_{\pm 0.4}$ |
| GroupDRO | | $59.5_{\pm 0.3}$ | | $24.0_{\pm 0.2}$ | | $62.2_{\pm 0.4}$ | | $64.6_{\pm 0.3}$ |
| Logits adjustment | | $77.5_{\pm 0.5}$ | | $22.9_{\pm 0.2}$ | | $77.6_{\pm 0.5}$ | | $64.0_{\pm 0.5}$ |
| Undersample | | $\underline{57.7}_{\pm 0.5}$ | | $\underline{21.6}_{\pm 0.3}$ | | $\underline{59.1}_{\pm 0.2}$ | | $\underline{57.2}_{\pm 0.3}$ |
| Oversample | | $59.7_{\pm 0.3}$ | | $24.1_{\pm 0.2}$ | | $62.8_{\pm 0.5}$ | | $63.5_{\pm 0.7}$ |
| OOD-Chameleon | | $\mathbf{55.8}_{\pm 0.4}$ | | $\mathbf{20.7}_{\pm 0.2}$ | | $\mathbf{58.3}_{\pm 0.2}$ | | $\mathbf{56.6}_{\pm 0.2}$ |

# I. Additional Results on Efficiency Considerations

The main point of training an algorithm selector is to **amortize its cost in the long run**. I.e. the selector is trained once to generalize to unseen datasets and shifts, as addressed in our experiments.

### I.1. Comparison with Ensemble, A Baseline with Much Higher Cost

In the paper, we compared baselines with similar costs to solve a downstream task. We now additionally compare our method with an additional baseline of uniformly ensembling multiple methods' predictions, whose computational requirement is multiple times higher than our method for each new downstream task. Our method performs significantly better than this much more expensive one, as shown in Table 19.

*Table 19.* Comparison between OOD-Chameleon and Ensemble.

|  | CelebA (ResNet) | CelebA (CLIP) | MetaShift (ResNet) | MetaShift (CLIP) |
|---|---|---|---|---|
| Uniform ensemble | 50.8 | 42.6 | 41.1 | 29.6 |
| OOD-Chameleon | **47.7** | **39.1** | **39.0** | **27.2** |

### I.2. Generalizing from Small Datasets to Larger Ones

One potential of the algorithm selector is training on datasets with smaller data sizes and then using it to select suitable algorithms on larger datasets at test time, which could further amortize the cost. Here we provide preliminary experiments

on this front. The algorithm selector is now trained with datasets of size smaller than or equal to 1k and used to select algorithms for datasets of size 2k and 3k. As shown in Table 20, we see that the algorithm selector still accurately predicts suitable algorithms.

*Table 20.* Performance on CelebA with varying data sizes.

| Data size | CelebA (ResNet) | CelebA (CLIP) |
|---|---|---|
| up to 1000 (i.e. results in Table 2) | 75.0 | 78.5 |
| 2000 | 81.7 | 83.8 |
| 3000 | 80.4 | 79.3 |

This concurs with our hypothesis that, by training on a meta-dataset of datasets with a range of smaller sizes, the model can learn to generalize to larger datasets by identifying patterns of the algorithms' performance w.r.t data sizes. We already showed (in Figure 3-right) that the variability in dataset size during training was critical.

## J. Additional Details on the Decision Trees

Here we include more analysis when the algorithm selector is implemented by decision trees, which is omitted in the main paper due to space constraints. In Figure 8 and Figure 9, we show the decision rules implemented by a (depth-3) decision tree when directly training on the CelebA meta-dataset. Since the problem is a binary multi-label classification task, the 'value' part is a list of size $(5, 2)$, and the rows correspond to 'ERM', 'GroupDRO', 'OverSample', 'Logits adjustment', 'UnderSample', respectively. One point that is not shown in the figures is that the final predictions are achieved by selecting the algorithm with the largest logit/probability (as mentioned in Section 2.2) when several of them are favorable. From the two figures, we see that different pieces of the descriptor are important in different cases: when ResNet-18 is used to run the algorithms, data size $n$, degree of label shifts $d_{ls}$ and the availability $r$ are important to inform the algorithm selection; while the degree of spurious correlation $d_{sc}$, degree of label shifts $d_{ls}$ and the availability $r$ are important in the case of CLIP.

**Decision trees trained to mimic (the best-performing) MLP algorithm selectors.** To understand what has been learned by the MLP-parametrized algorithm selector, we also investigate training a decision tree to directly mimic the function implemented by the MLP. *In our experiments, it gives a similar performance to the previous version of the decision tree, but it has the advantage of understanding the decision rule as a standard multi-class classification (hence the decision rules are more understandable).* Specifically, we obtain the predictions of the trained MLP (the predictions are one-hot since we pick the algorithm with the highest predicted logit/probability, as mentioned in Section 2.2) on the meta-dataset and use them to train a decision tree. To make the trees more interpretable, we convert the degree of shift by $|d_{(\cdot)} - 0.5|$ (where $|\cdot|$ is the absolute value) since by design the strength of shift is only identified by how far it is from 0.5 (balanced). In Figure 10 and Figure 11, we show the learned trees. We can see clearly how the different data characteristics determine the choice of the best algorithms. See Section 4.5 on more discussions on these two figures.

Overall, these decision rules of the tree can further help practitioners understand the applicability of the existing algorithms.

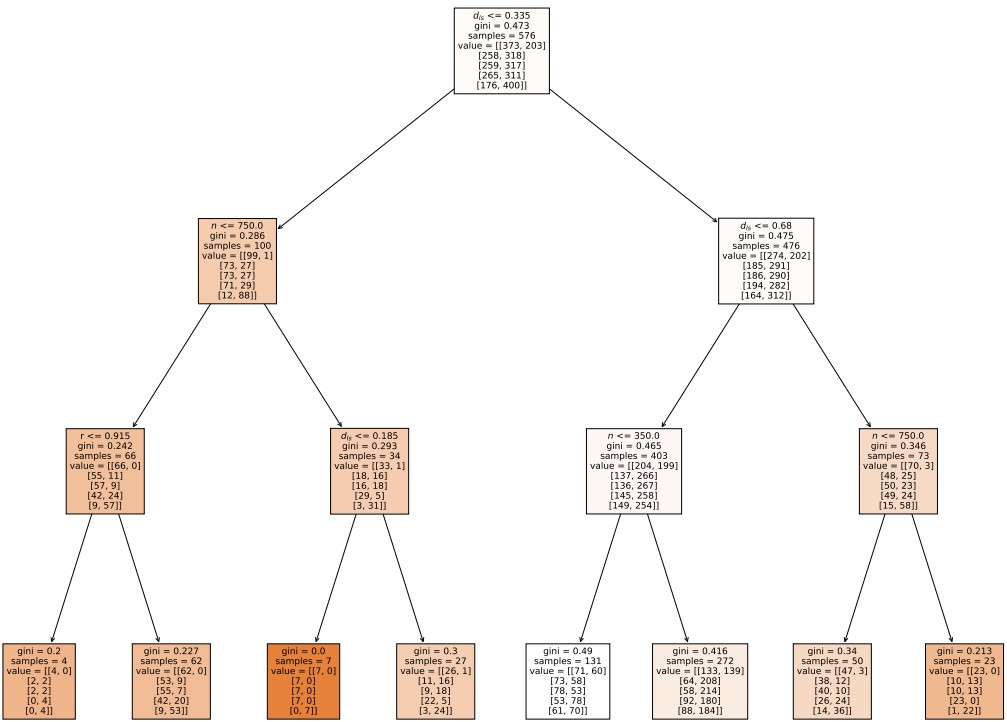

*Figure 8.* Visualization of a **algorithm selector implemented as a decision tree**, trained on the **CelebA** meta-dataset with **ResNet18** models.

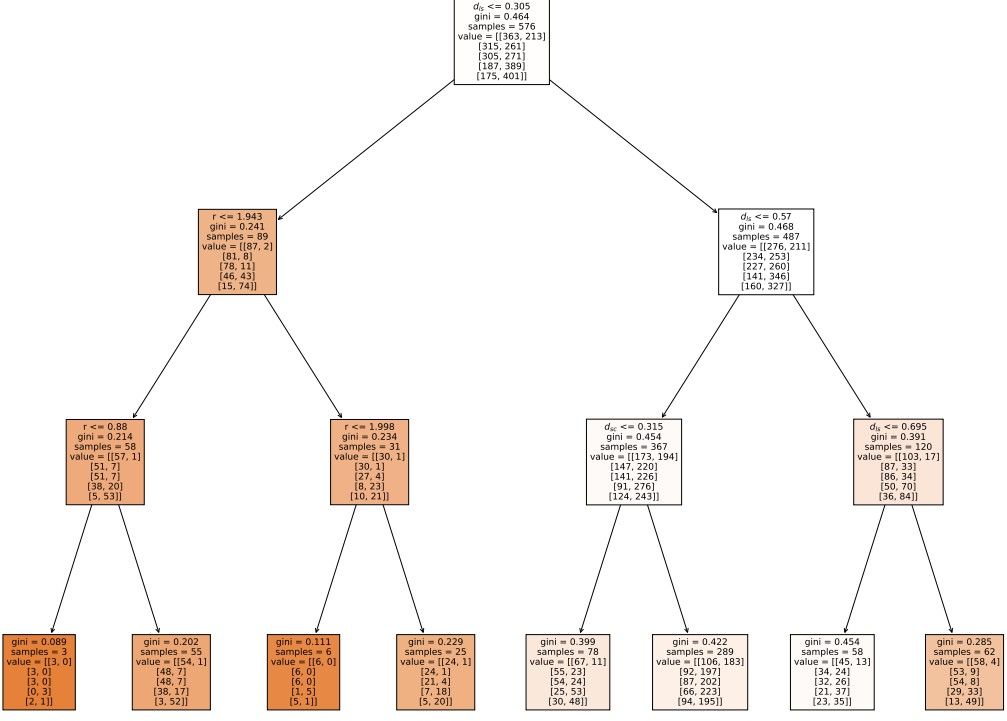

*Figure 9.* Same as Figure 8, with **CLIP (ViT-B/32)** architectures.

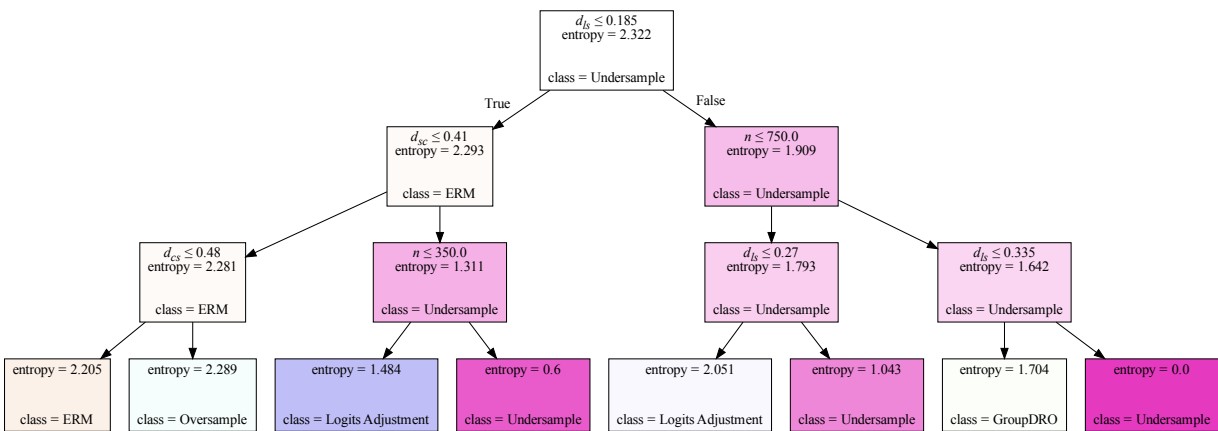

*Figure 10.* Visualization of a decision tree algorithm selector trained to mimic the one-hot predictions (obtained by selecting the top logits) of the best-performing (MLP) algorithm selector obtained from **CelebA** meta-dataset with **ResNet18** models. Since it is trained on one-hot labels, this version of the tree thus has the advantage of understanding the decision rule as a standard multi-class classification. To make the decision rule even more interpretable, we convert degrees of shift with $|d_{(\cdot)} - 0.5|$ where $|\cdot|$ is the absolute value. This accounts for the fact that, by design, the strength of shift is only identified by how far it is from 0.5 (balanced). See Figure 4 for a simplified version of this tree.

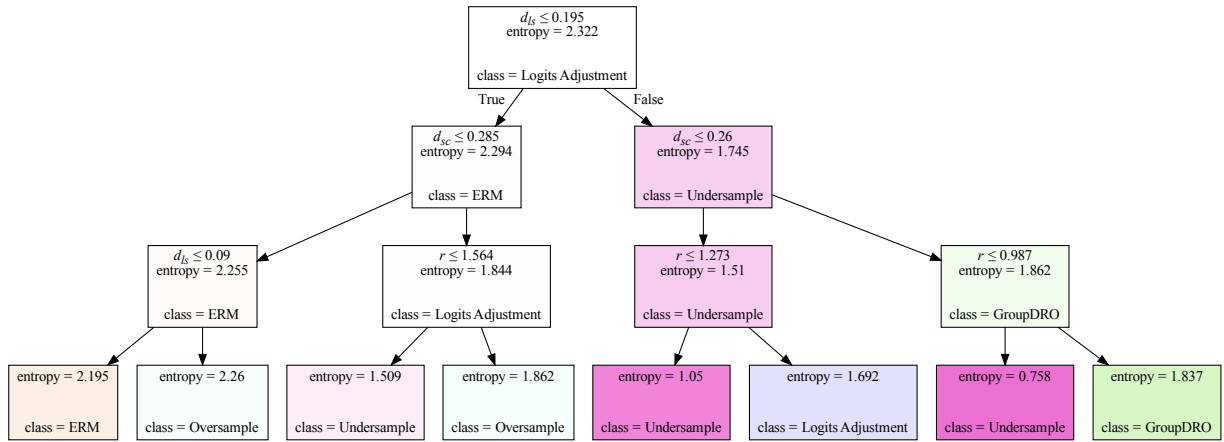

*Figure 11.* Same as Figure 10, with **CLIP (ViT-B/32)** architectures.

