# OpenReview forum: "OOD-Chameleon: Is Algorithm Selection for OOD Generalization Learnable?"
_ICML.cc/2025/Conference — ICML 2025 poster_

### Official Review · Reviewer_FvV1 · 2025-02-17

**Overall Recommendation:** 2

**Summary:**

The paper introduces OOD-Chameleon, a framework for automatically selecting training algorithms to improve out-of-distribution (OOD) generalization. Instead of trial-and-error, it learns to predict the best algorithm based on dataset characteristics. The authors create a meta-dataset by re-sampling datasets with various distribution shifts and train a model to classify which algorithms (e.g., ERM, GroupDRO, oversampling) perform best. Experiments in vision and language tasks show that OOD-Chameleon outperforms static algorithm selection and learns interpretable decision rules, demonstrating that algorithm selection for OOD generalization is learnable and effective.

**Claims And Evidence:**

Yes.

**Essential References Not Discussed:**

No.

**Experimental Designs Or Analyses:**

Yes.

**Methods And Evaluation Criteria:**

Yes.

**Other Comments Or Suggestions:**

Comments:
1. Lack of Theoretical Guarantees – The paper primarily relies on empirical experiments but lacks a clear theoretical analysis to explain why OOD-Chameleon can successfully predict the optimal algorithm.
2. Although decision tree analysis provides some insights, the study does not deeply discuss the problem from the perspective of generalization bounds or an OOD theoretical framework.
3. The paper compares simple baselines (ERM, GroupDRO, Oversampling), but it may lack stronger relevant methods as additional baselines.
4. The method relies on a meta-dataset, but if the meta-dataset lacks diversity, the performance may degrade on new tasks.

**Other Strengths And Weaknesses:**

Strengths:
1. This study proposes a data-driven algorithm selection method, rather than relying on traditional trial-and-error strategies. The idea of automatically selecting training algorithms is novel in the field of OOD generalization.
2. The paper provides some interpretability through decision tree analysis, examining how different algorithms perform under various dataset characteristics, which enhances the credibility of the study.

Weaknesses:
See comments.

**Questions For Authors:**

Questions:
1. If the shift types in the training data and test data are completely different, can the selector still work effectively?
2. Test the performance of this method in more complex OOD scenarios, such as the Office-Home dataset.

**Relation To Broader Scientific Literature:**

No.

**Theoretical Claims:**

Yes.

---

> ### Author Rebuttal · Authors · 2025-03-31
>
> Many thanks for the thoughtful review. We are happy that you found our work novel and the analysis insightful.
>
> ---
>
> **`Comment 1 (Theoretical analysis to explain the success of OOD-Chameleon):`** The bottom line is that our meta-learning-like approach turns algorithm selection into a *supervised* problem where, on the theoretical side, classical results from statistical learning theory apply. In particular, we use either an MLP or a decision tree to parametrize the algorithm selector. The approximated VC dimensions of these classes are well defined and can be used for bounding generalization errors of the selector.
>
> ---
>
> **`Comment 2 (Theoretical analysis from an OOD framework):`** This is a valid question. Identifying generalization bounds when multiple types of shifts co-exist is an open problem. Most theoretical work focuses on a given type of distribution shift [1,2,3]. **The difficulty of theoretically handling complex mixed types of shifts is a core motivation** for our empirical approach (see L40, L66, L185). A theoretical treatment would be highly valuable but besides the point of this paper and deserves a full separate work.
>
> [1] Understanding the Failure Modes of Out-of-Distribution Generalization. Nagarajan et al.
>
> [2] A Unified View of Label Shift Estimation. Garg et al.
>
> [3] A Theoretical Analysis on Independence-driven Importance Weighting for Covariate-shift Generalization. Xu et al.
>
> ---
>
> **`Comment 3 (additional algorithms/baselines):`** Our approach is **not** tied to specific algorithms. The ones used in the current implementation are justified L245-255: (1) they belong to different categories, (2) can address different types of shifts, (3) were proven to be effective in seminal works. Most importantly, no method is **strictly stronger** than these [4,5], and we show that *better algorithm selection with these 'simple' methods is sufficient to improve OOD generalization*.
>
> [4] Simple data balancing achieves competitive worst-group-accuracy, Idrissi et al.
>
> [5] Change is Hard: A Closer Look at Subpopulation Shift, Yang et al.
>
> **Algorithm selection baselines:** There are no existing competing methods; our evaluation includes various single-algorithm selection baselines. We propose an additional experiment with a new strong **ensemble** baseline: every algorithm is used to train a model whose predictions are ensembled (**row 1**). Our approach (**row 2**) significantly outperforms this ensemble. The ensemble also requires training and evaluating as many models as algorithms vs. only one for our method.
>
>
> | (Worst-group error, lower is better) |CelebA (ResNet)|CelebA (CLIP)| MetaShift (ResNet) | MetaShift (CLIP) |
> |-|-|-|-|-|
> |Uniform ensemble (new)|50.8|42.6|41.1|29.6|
> |OOD-Chameleon (from the paper)|**47.7**|39.1|**39.0**|27.2|
> |+ Weighted ensemble (new)|48.3|**38.9**|39.2|**26.8**|
>
> As suggested by **Reviewer sRjw**, we also investigated combining OOD-Chameleon with an ensemble (**row 3**): predictions are ensembled after weighting by the predicted score of the algorithm. This further improves OOD-Chameleon in certain cases. It confirms the quality of the predicted suitability of algorithms, but removes the benefit of only having to train a single model.
>
> ---
> **`Comment 4 (meta-dataset diversity):`** This is true for any supervised model where the training data coverage will inevitably affect the performance (also similarly, the pretraining data coverage for a pre-trained model affects its capability). So we rather believe this is a feature instead of a limitation, since this allows for building a more powerful predictor with meta-dataset scaled up. This is confirmed in Figure 3-middle. Another point to note in that figure is that, only a *relatively small size* of meta-dataset (~200) is sufficient to outperform the best non-parametrized baseline by a large margin.
>
> ---
> **`Q1`**: please see the response to **Reviewer sPTX** (**Q4**).
>
> ---
> **`Q2 (Office-Home):`** We performed additional experiments as requested with Office-Home. We created 100 datasets by randomly sampling a pair of domains and a pair of classes for each dataset (e.g. classifying pen/knife in Art/Clipart). The number of samples is naturally imbalanced, creating mixed types of distribution shifts by design. We use the algorithm selector from the paper trained with the "CelebA" meta-dataset, which we use to predict suitable algorithms for these variants of Office-Home. The (worst group error) results are below. They provide strong additional support for the utility of our approach to select suitable algorithms across datasets and diverse types of shifts.
>
>
> || Office-Home (ResNet) | Office-Home (CLIP) |
> | - | - | - |
> | Oracle selection | (14.8) | (11.4) |
> | Random selection | 19.3 | 15.1 |
> | Global best | 18.2 | 14.4 |
> | Naive descriptors | 18.8 | 14.6 |
> | Regression | 18.5 | 14.3 |
> | OOD-Chameleon | **17.9** | **13.5** |
> ---
> Let us know if there are remaining questions or weaknesses that could be addressed. Thanks!

---

### Official Review · Reviewer_9QSc · 2025-02-24

**Overall Recommendation:** 4

**Summary:**

This paper presents _OOD-Chameleon_, a principled and bottom-up approach for selecting training algorithms in out-of-distribution (OOD) generalization tasks. By framing algorithm selection as a multi-label classification problem, the method leverages a meta-dataset of datasets representing diverse distribution shifts to select algorithms based on test data descriptors. The key contribution is enabling zero-shot algorithm selection by leveraging related tasks/datasets, eliminating the need for grid-searching over algorithmic choices. The method is evaluated across synthetic, vision, and language tasks, demonstrating strong performance in identifying high-performing algorithms for unseen datasets.

**Strengths**
- **Addresses a timely problem:** No single algorithm consistently outperforms empirical risk minimization (ERM) across diverse OOD settings, making dataset-aware and automated algorithm selection valuable.
- **Strong empirical evaluation:** The model achieves near-oracle performance across multiple domains (synthetic, vision, NLP) and generalizes to new datasets, outperforming multiple representative baselines.
- **Interesting analysis:** The learned selector captures non-trivial dataset-algorithm relationships and provides analysis on when different interventions are effective.
- **Clean and simple framework:** The multi-label classification approach is intuitive and extensible, supported by thorough ablation studies over model class (linear, k-NN, decision trees) and cost function (multi-label vs. regression).

**Weaknesses**
- **High upfront cost:** Constructing the meta-dataset requires significant computational resources, as it involves training models across many datasets and configurations. It is unclear how well this would scale to larger-scale settings, or whether a predictor trained on a metadataset of small datasets would generalize to a larger dataset at test time
- **Generalization to unseen shifts:** This (supervised learning) method should perform well when test datasets resemble those in the meta-dataset, but its ability to generalize under significant distribution shifts is unclear. Careful curation of the meta-dataset and dataset descriptors feature engineering is required, which in turn requires some domain knowledge about the test datasets. The paper should explicitly discuss this.
- **Lack of evaluation on larger-scale algorithmic selection tasks:** While the framework is general, experiments primarily focus on semi-synthetic datasets (e.g., CelebA, MultiNLI) and worst-group accuracy tasks. Evaluating the method on large-scale OOD problems—such as language model pretraining or ImageNet-scale classification—would better demonstrate its broader applicability. Even analyzing how basic hyperparameters (e.g., optimizer settings, scheduler choices, vocabulary size) and pretraining dataset composition affect final model performance would be interesting.
- **No compute-matched baseline:** The study lacks a direct comparison against random grid search using equivalent compute resources (i.e., the cost of constructing the meta-dataset). Additionally, potential synergies with search-based approaches are unexplored—for instance, using OOD-Chameleon’s predicted algorithm selection to warm-start a grid search over the test dataset.

**Questions & Suggestions:**
- How does the method handle algorithms that require different levels of metadata? Could this bias the predictor toward metadata-rich approaches (e.g., GroupDRO vs. ERM)?
- The section on quantifying distribution shifts (Section 3) is difficult to follow. The heuristics for measuring shifts should be explained more clearly.

**Claims And Evidence:**

Yes

**Essential References Not Discussed:**

None that I know of.

**Experimental Designs Or Analyses:**

Yes

**Methods And Evaluation Criteria:**

Yes

**Other Comments Or Suggestions:**

Please see my review

**Other Strengths And Weaknesses:**

Please see my review

**Questions For Authors:**

Please see my review

**Relation To Broader Scientific Literature:**

A new dataset-aware zero-shot method for algorithm selection

**Theoretical Claims:**

N/A

---

> ### Author Rebuttal · Authors · 2025-03-31
>
> Thanks for the constructive feedback. We are glad that you found our work timely, valuable, and appreciated its clarity and insights.
>
> ---
>
> **`W1 (Would a predictor trained on a meta-dataset of small datasets generalize?):`** We performed additional experiments, reported below. The algorithm selector is now trained with datasets of size <= 1k and used to select algorithms for datasets of size 2k and 3k. We see that the algorithm selector still accurately predicts suitable algorithms.
>
> This concurs with our hypothesis that, by training on meta-dataset of datasets with *a range of smaller sizes*, the model can learn to generalize to larger datasets by identifying patterns of the algorithms' performance w.r.t data sizes. We already showed (Figure 3-right) that the variability in dataset size during training was critical.
>
> | Data size | CelebA (ResNet) | CelebA (CLIP) |
> | -- | -| -|
> | up to 1000 (i.e. Table 2) |  75.0    | 78.5     |
> | 2000     | 81.7     | 83.8     |
> | 3000     | 80.4     | 79.3     |
>
> (Numbers are algorithm-selection accuracy; higher is better)
>
> ---
>
> **`W2 (Generalization to unseen shifts):`** Because of space limits, we refer to the response to **Reviewer sPTX** (**Q4**). We are adding an extended discussion of this important point to the paper.
>
> ---
>
> **`W3 (large-scale evaluation):`** Agreed, all of these points are important considerations that deserve an entire dedicated publication. Automating OOD algorithm selection is a completely new take on OOD generalization, and the point of this study (as stated in the title) is to investigate the *feasibility* of such a radically different approach.
>
> We concur that these are all very exciting extensions, including automating the extraction of useful dataset characteristics, and we are actively working on it.
>
> ---
>
> **`W4 (compute-matched baseline):`** It is unclear what would be the amount of compute to match: we clarified in the paper that the whole point of training an algorithm selector is **to amortize its cost in the long run**. I.e. the selector is trained once to generalize to unseen datasets and shifts, as addressed in our experiments.
>
> **Comparison with ensemble, a baseline with much higher cost.** In the paper, we compared baselines with similar costs to solve a downstream task. To address the proposed suggestion, we now additionally compare our method with an additional baseline of ensembling multiple methods predictions, whose compute requirement is multiple times higher than our method for each new downstream task. For space reasons, these new results are reported in the response to **Reviewer FvV1** (**Comment 3**). Our method performs significantly better than this much more expensive one.
>
> Regarding your comment on 'synergies', we additionally explored the combination of our method with ensemble-based methods -- using the predicted algorithm selection to adjust ensemble weights. We found that this strategy further improves in certain cases, albeit with a higher cost. Please again refer to the response to **Reviewer FvV1** (**Comment 3**) for more details.
>
> ---
>
> **`Q1 (different levels of meta-data):`** The results indiate that the predictor does **not** have such as bias. For example, in Figure 3-left, the comparison between ERM/Undersample (orange) shows that ERM outperforms Undersample in many cases even though the latter uses meta-data (i.e. spurious attribute labels). Prior work [1,2,3] already pointed out that ERM can outperform complex methods that use rich meta-data, which is part of the underlying motivation for our work.
>
> [1] OoD-Bench: Quantifying and Understanding Two Dimensions of Out-of-Distribution Generalization, Ye et al.
>
> [2] In Search of Lost Domain Generalization, Gulrajani et al.
>
> [3] MetaShift: A Dataset of Datasets for Evaluating Contextual Distribution Shifts and Training Conflicts, Liang et al.
>
> ---
>
> **`Q2 (measuring distribution shifts):`** We reworked this section, which should now be much clearer. In summary, for each sample in the training data of a dataset, we have class label and attribute (or domain) label annotated. Then:
> - The bias of class label distribution indicates the degree of label shifts.
> - The bias of attribute (or domain) label distribution indicates the degree of covariate shifts.
> - The correlation of class labels and attribute (or domain) labels indicates the degree of spurious correlation.
> - Why is it sufficient to use only training data to measure the distribution shift? Because we use an unbiased test set, then how biased a distribution is (to an unbiased one) directly reflects the degree of shifts.
>
> ---
>
> Let us know if there are remaining questions or weaknesses that we could address. Thanks again!

---

### Official Review · Reviewer_sPTX · 2025-03-12

**Overall Recommendation:** 3

**Summary:**

This paper introduces an interesting task: choosing the right training algorithm for the right dataset. The authors propose an approach called OOD-CHAMELEON, which predicts whether a given algorithm can generalize to unseen OOD data based on its past performance on the current dataset. I find this work sufficiently novel and engaging, as adapting existing algorithms to different OOD scenarios may be more practical than designing entirely new ones. However, some aspects of the paper lack clarity in detail. If the authors can address my concerns, I would be willing to reconsider my score.

**Claims And Evidence:**

yes

**Essential References Not Discussed:**

yes

**Experimental Designs Or Analyses:**

yes

**Methods And Evaluation Criteria:**

yes

**Other Comments Or Suggestions:**

see Questions

**Other Strengths And Weaknesses:**

Strengths :

1. interesting topic, and I think utilize Algorithm Selection to solve OOD problem is novel.
2. This paper is well-organized and easy to read.
3. Complete experimental instructions

Weaknesses:

see Questions

**Questions For Authors:**

1. If the method follows a multi-label classification approach as in Equation (3), how is the most effective algorithm selected during the testing phase? While I understand the authors' claim that "multiple algorithms can sometimes be equally suitable," this reasoning does not seem applicable to the testing stage.

2. Given that the task is multi-label classification, why does Equation (3) use cross-entropy (CE) instead of binary cross-entropy (BCE)?

3. Is the dataset descriptor represented as an indicator vector? Although the authors provide a computational method in the appendix, I still find its definition unclear.

4. The authors claim that OOD-CHAMELEON is a data-driven performance prediction model, which intuitively means leveraging past performance to infer future performance. While I accept this assumption when the datasets used for performance evaluation remain unchanged, how can this assumption still hold in an OOD setting? I am concerned about cases where an algorithm performs poorly on the in-distribution (ID) dataset but is actually well-suited for handling unseen OOD data.

**Relation To Broader Scientific Literature:**

They try to connect algorithm selection and OOD generalization

**Theoretical Claims:**

no theory in this paper

---

> ### Author Rebuttal · Authors · 2025-03-31
>
> Thanks for the thoughtful feedback. We are glad that you found our work novel and well presented.
>
> ---
>
> **`Q1 (algorithm selection at test time):`** We simply pick the method with the *highest logit* (L159 right column), which corresponds to the one the model is the most confident in.
>
> An alternative would be to use binary predictions, and if multiple algorithms are predicted to be suitable, randomly select one of them. Empirically, the former option (top logits) is much better. See below for new experiments comparing these options (worst-group error is shown, lower is better, and experimental setup is the same as Table 2 in the paper).
>
> | Selection criterion | CelebA (ResNet) | CelebA (CLIP) | MetaShift (ResNet) | MetaShift (CLIP) |
> | -- | -- | -- | -- | -- |
> | Top logits (as in the paper)         | 47.7 | 39.1 | 39.0 | 27.2 |
> | Binary predictions (new experiments) | 48.5 | 39.8 | 39.4 | 27.4 |
>
> ---
> **`Q2 (loss function):`** Yes it uses the **binary** cross-entropy, we will make this clearer.
>
> ---
> **`Q3 (dataset descriptor):`** Each component in the vector is either a real number (e.g. the degree of distribution shifts or the availability of spurious feature) or an integer (e.g. size of the dataset). We will make the definition clearer.
>
> ---
>
> **`Q4:`** The answer is subtle but absolutely critical to understanding the value of our approach.
> The key innovation is (1) to lift the challenge from OOD generalization across specific shifts, to a meta level of generalizing across **distributions of distribution shifts**. And (2) to show that we can automatically sample semi-synthetic datasets (to train an algorithm selector) after choosing a realistic distribution of distribution shifts.
>
> This is conceptually *a completely novel approach to OOD generalization*. While it is true (*for any learning method!*) that there is no guarantee of generalizing beyond the support of the training distribution, the training distribution in our approach encompasses a wide variety of shift types, magnitudes and their combinations, which can be densely sampled (point (2) above). We acknowledge that new shift types might require additional curation (we only claim that our approach should work a distribution of distribution shifts, as noted in section 2.1). We propose to state more clearly our assumptions and limitations to avoid any overclaiming.
>
> **OOD generalization on data sizes**: However, we hypothesize that it is possible to generalize to OOD data size -- the algorithm selector has learned to generalize to larger data sizes, potentially by exploring the different data sizes available. In our response to **Reviewer 9QSc** (**W1**), we show that this is indeed the case. This is a case where the algorithm selector generalizes to OOD data.
>
> ---
> Let us know if there are remaining questions or weaknesses that we could address. Thanks again!

---

### Official Review · Reviewer_sRjw · 2025-03-12

**Overall Recommendation:** 3

**Summary:**

This paper tackles the interesting problem of algorithm selection for OOD generalization. In particular, the paper proposes a meta-learning-like algorithm which takes in summary statistics from a dataset as well as a distributionally robust learning algorithm. This tuple is then mapped to the performance of training said algorithm on the dataset described by the summary statistics. For a new dataset, this mapping can be used to predict the likely performance of various distributionally robust learning algorithms without actually training these models. This allows for a flexible algorithm choice across a wide set of shifts and spurious correlations. The effectiveness of the method is demonstrated empirically on synthetic, image, and text datasets. Further, the paper provides general insights on the conditions under which the benchmarked distributionally robust learning algorithms perform well.

### Update after rebuttal

I thank the authors for their rebuttal. I think my concerns have been addressed sufficiently to warrant a weak accept. The connection to DRO still deserves further investigation, though, which is why I don't see myself in a position to champion the paper.

**Claims And Evidence:**

See next section.

**Essential References Not Discussed:**

Prior work appears to be sufficiently discussed.

**Experimental Designs Or Analyses:**

In general, I think that the experimental design makes sense and is similar to other works in the meta learning community. My biggest concern here is the lack of clarity in Section 3 (see comments below in strengths and weaknesses).

**Methods And Evaluation Criteria:**

In principle, the method makes sense and should work *under the assumption that the distribution of dataset descriptors is representative of descriptors encountered at deployment time*. I don't think that this assumption is explicitly acknowledged in the paper, though. While it seems like the empirical results show some degree of generalization across datasets and tasks, I am skeptical whether this will necessarily translate in general. Since the learning process described in 2.2 is standard empirical risk minimization and not a distributionally robust algorithm itself, there is no guarantee that this generalization pattern should necessarily hold in general beyond the support of the training distribution. I believe that the paper is over-claiming the generalization capabilities of the proposed method. Also, as discussed in my questions below, it would be worthwhile to report a broader notion of error beyond just the worst group. Evaluating distributional generalization methods solely using worst-group error risks overfitting to small or narrowly defined groups, causing instability and obscuring performance trade-offs with other critical groups.

**Other Comments Or Suggestions:**

- The color bars in the tables are a great way to show the algorithm allocation. Big fan of this visualization!
- Section 3: Should Figure 3 be Figure 2?
- Lines 156 (right column): I assume you use *binary* cross entropy here? This would be worth making explicit to avoid confusion.

**Other Strengths And Weaknesses:**

**Strengths**: The question posed by the paper---is it possible to select a distributionally robust learning method based on properties of a dataset---is interesting and relevant to the reliability community. The paper also includes experiments across data modalities (synthetic, image, language). I also find the insights from section 4.5 on the task <---> method fit interesting.

**Weaknesses**: I think Section 3 is currently not very clear. In particular, it is unclear to me how the covariate shifts are generated and what degree of variability can be achieved. While label shifts and spurious correlations are a bit more limited in terms of their simulation flexibility, covariate shifts can happen in many different forms. I don't find this sufficiently discussed in the current draft which makes me question the method's effectiveness in the downstream experimental results. As noted by the authors themselves, the fact that the dataset descriptor is designed with engineered features constitutes a limitation that is likely to affect the proposed method. In particular, key dataset properties are often not directly observable but need to be estimated from the data. While the paper talks about estimating spurious correlations, it does not seem to discuss techniques for estimating the presence and intensity of covariate and label shifts. The effectiveness of the method could be strongly impacted by the quality of this estimation.

**Questions For Authors:**

1) Lines 91-92 (right column): "But distribution shifts appearing in real-world data are not arbitrary." I am not sure what is meant here by "arbitrary". Can you clarify?
2) Line 98 (right column): Can you introduce some foreshadowing examples here already on the dataset characteristics?
3) Lines 113-116: I agree that worst group error is a key metric to consider but would it be possible to report a broader notion of error beyond just the worst group? Evaluating distributional generalization methods solely using worst-group error risks overfitting to small or narrowly defined groups, causing instability and obscuring performance trade-offs with other critical groups.
4) Lines 123-124: I am not sure this is true. In most practical applications where shifts occur (and where OOD generalization is therefore important), directly observing labels for spurious attributes is prohibitive. Can you clarify this statement?
5) What is the method's sensitivity to the suitability hyperparameter $\epsilon$?
6) Why do you compare pairs of methods in Figure 3 (left)? I am not sure what this figure is telling me.
7) Is oracle selection simply done by running all algorithms on all tasks and then picking the best model across all tries?
8) Line 216: $\mathcal{G}_i$ has never been introduced before. What is this?
9) Can you comment on the applicability of ensembling to the problem you raise? Would it be possible to train all methods and then use a weighted average to generate predictions?

**Relation To Broader Scientific Literature:**

The related work section sufficiently discusses the role this work plays in the context of OOD generalization, algorithm selection, and meta learning.

**Theoretical Claims:**

The paper does not provide any theoretical claims. It would be great to gain some more theoretical insights from learning theory to understand under what conditions a generalizable algorithm predictor is learnable. Some connections to meta learning could probably be useful here.

---

> ### Author Rebuttal · Authors · 2025-03-31
>
> Thanks for the insightful review. We are glad that you found the studied problem and insights interesting. We address the questions below and propose several improvements to the paper.
>
> ---
> **`W1: No guarantee beyond the support of the training distribution:`**
>
> Correct, this is true of any learning method. The subtle but important innovation is (1) to lift the challenge from OOD generalization across *specific* shifts, to a meta level of generalizing across **distributions of distribution shifts**. And (2) to show that we can automatically **densely sample** semi-synthetic datasets (to train an algorithm selector) **within** a distribution of realistic distribution shifts.
>
> This is a conceptually novel approach to OOD generalization. We propose to state more clearly our assumptions and limitations to avoid any overclaiming.
>
> Please also see the response to **Reviewer FvV1** (**Comment 1/2**) regarding theoretical claims.
>
> ---
> **`W2: Variety of covariate shifts:`** Thanks for pointing this out, we clarified section 3. In summary, we simulate changes in $P(X)$ by intervening on the domain distribution. Covariate shifts can indeed happen in many ways, but this simple approach is both tractable and generates realistic shifts (e.g. with image data, introducing more from an *indoor* domain in one dataset vs. more from *outdoor* in another dataset).
>
> To simulate different degrees of covariate shifts, we draw samples with different distributions of attribute or domain labels. The variability covers the full range with a ratio of samples from any domain in $[0,1]$ (thus including severe shifts).
>
> Please also see the response to **Reviewer 9QSc** (**Q2**) for a general view of section 3.
>
> ---
> **`W3: Techniques for estimating covariate/label shifts:`**
>
> We added to the paper a discussion of the literature devoted to this (e.g. [1]). To clarify, the estimation method we use applies to both label and covariate shifts: all it does is to infer sample-wise domain labels then use them to compute the estimated degrees of shifts.
>
> [1] Unified View of Label Shift Estimation, Garg et al.
>
> ---
> **`Q1 (arbitrary shifts):`** We will expand and clarify: if shifts were arbitrary, inductive reasoning and machine learning would not be possible [2,3] and the proposed approach also could not help. We only claim the possibility of training an algorithm selector that is effective on a distribution of distribution shifts, not on *any* possible shift.
>
> [2] The Lack of A Priori Distinctions between Learning Algorithms, Wolpert
>
> [3] Domain Adaptation under Target and Conditional Shift. Zhang et al.
>
> ---
> **`Q2:`** Great suggestion - done.
>
> ---
> **`Q3 (worst-group error):`** Agreed. We repeated our experiments and now additionally report the **averaged-group error**. The conclusions are similar to those in the paper.
>
> (The experimental setup follows Table 2 for all new results.)
> | |CelebA (ResNet)| CelebA (CLIP) | MetaShift (ResNet) | MetaShift (CLIP) |
> | - | - | - | - | - |
> |Oracle selection|(33.8)|(26.4)|(24.6)|(14.8)|
> |Random selection|36.6|29.5|27.8|18.1|
> |Global best|35.7|28.0|26.9|16.6|
> |Naive descriptors|35.8|27.7|26.7|16.9|
> |Regression|35.5|28.3|27.0|16.4|
> |OOD-Chameleon|**35.1**|**27.2**|**26.3**|**15.9**|
>
> ---
> **`Q4 (spurious attributes):`** Our main experiments follow the setup from much of prior work on OOD generalization (domain generalization, spurious correlation, e.g. [4,5], and many more) which indeed assumes access to attribute labels. We also evaluate scenarios without such labels (Appendix E).
>
> [4] In Search of Lost Domain Generalization, Gulrajani et al.
>
> [5] Change is Hard: A Closer Look at Subpopulation Shift, Yang et al.
>
> ---
> **`Q5 ($\epsilon$):`** We performed additional experiments. The method proves quite robust to other choices than the $\epsilon=0.05$ used in the paper (bold row). Intuitively, when $\epsilon$ is too big, the meta-dataset loses discriminative power; too small, it is subject to noise.
>
> | $\epsilon$ | CelebA (ResNet) | CelebA (CLIP) | MetaShift (ResNet) | MetaShift (CLIP)|
> | - | - | - | - | - |
> |0.0  | 48.3 | 39.8 | 39.9 | 27.7 |
> |0.025| 48.1 | 39.6 | 39.7 | 27.3 |
> |0.05 | **47.7**| **39.1**| **39.0**| **27.2**|
> |0.10 | 48.0 | 39.4 | 39.6 | 27.8 |
>
> ---
> **`Q6 (comparison of pairs):`** This highlights the discrepancy in performance among methods under different shifts. Without this, there would be nothing to learn. E.g. the comparison of under-/oversampling (blue) shows that many datasets benefit specifically
>  from one *or* the other.
>
> ---
> **`Q7 (oracle selection):`** Yes.
>
> ---
> **`Q8 ($\mathcal{G}_i$):`** Number of samples in group $i$ (defined L114, illustrated in Figure 2). We will make it clearer.
>
> ---
> **`Q9 (ensembling):`** It is indeed possible. We tested two ensemble methods in response to **Reviewer FcV1** (**Comment 3**).
>
> ---
> We incorporated the above clarifications and improvements into the manuscript. Let us know if there are remaining questions or weaknesses that we could address. Thanks!

---

> > ### Comment · Reviewer_sRjw · 2025-04-02
> >
> > I thank the authors for the rebuttal which has clarified some of my questions. However, some concerns remain:
> >
> > > Correct, this is true of any learning method. The subtle but important innovation is (1) to lift the challenge from OOD generalization across specific shifts, to a meta level of generalizing across distributions of distribution shifts. And (2) to show that we can automatically densely sample semi-synthetic datasets (to train an algorithm selector) within a distribution of realistic distribution shifts.
> >
> > Can you explain how your approach relates to techniques in distributionally robust optimization (DRO)? Instead of optimizing just for the average case, DRO focuses on minimizing the worst-case loss over a set of plausible distributions. If your goal is to generalize across distributions of distribution shifts, then DRO sounds highly related.
> >
> > I also don't think that this concern was explicitly acknowledged in the rebuttal:
> >
> > > In principle, the method makes sense and should work under the assumption that the distribution of dataset descriptors is representative of descriptors encountered at deployment time. I don't think that this assumption is explicitly acknowledged in the paper, though.
> >
> > I understand that there are severe space constraints in the original response so it would be great if the authors could comment on this.

---

> > > ### Author Response · Authors · 2025-04-03
> > >
> > > Dear Reviewer sRjw,
> > >
> > > Thank you for your follow-up comments and for engaging critically with our work. We appreciate the opportunity to further clarify our work.
> > >
> > > ---
> > >
> > > **`Relation to DRO`**:
> > >
> > > You raised a great point about the connection to DRO. There are two perspectives:
> > >
> > > - On a given dataset, just like other (OOD generalization) algorithms, DRO can be suboptimal in certain cases. This is because, DRO can produce overly conservative solutions in many cases. Importantly, on the mixture of shifts scenarios we considered in this paper, defining DRO's ambiguity set is non-trivial [1]. These reasons make it prohibitive to *directly* apply DRO to generalize across the considered distribution of distribution shifts.
> > > - On a meta level, when training the algorithm selector, it is possible to replace the empirical risk minimization (ERM) with a DRO-like objective, for some robustness to *meta*-distribution shifts. We do not think this is needed though, because of the assumption that we will discuss in the response to your next question.
> > >
> > > [1] Distributionally Robust Neural Networks for Group Shifts, Sagawa et al.
> > >
> > > **Our action**: We will add a discussion devoted to the above (will be somewhere in Section 2.2). Moreover, the discussions with you inspired us to discuss the *interaction* of our framework with other approaches in the paper. For example, DRO can be applied to train the algorithm selector, and Weighted Ensemble can be performed on top of predictions from the algorithm selector.
> > >
> > > ---
> > > **`Representativeness of dataset descriptors at deployment time`**:
> > >
> > >
> > > Sorry for the confusion, we meant to explicitly acknowledge this assumption. The generalization ability of our approach indeed relies on the premise that the dataset descriptors used during training are representative of those encountered at deployment. While this is an inherent challenge for most, if not all (meta-)learning frameworks, we acknowledge that test-time shifts in these descriptors should impact the performance.
> > >
> > > **Our action**: We will add **explicit** discussions about this assumption to the paper:
> > > - In *Section 2.1 (Is the Selection of the Best Algorithm Possible?)*, we will frame this assumption formally as **Assumption 1** and clarify the surrounding text.
> > > - Then in *Section 6 (Conclusion)*, we will further discuss
> > >     1.  The potential limitations of this assumption (e.g. failure modes of generalizing to new types of shifts, such as other types of covariate shifts).
> > >     2.  What future work can do about it. For example, expanding metadataset diversity, etc.
> > >
> > > ---
> > > Hope this addresses your concerns and thus improves your assessment of our paper. Thanks again for the detailed review and for engaging in discussions, your comments have greatly improved our work.

---

### Decision · Program_Chairs · 2025-05-01

**Decision:**

Accept (poster)

**Comment:**

The reviewers generally agreed that the work merits acceptance. The paper addresses the timely issue of selecting OOD generalization methods where there is not a consistent method across different types of shifts, and shifts may be unknown in practice. Overall, the reviewers found their proposed meta-learning method using dataset properties to predict which OOD generalization will most likely result in good OOD performance. Ironically, there are some concerns about the OOD generalizability of the method and suggestions that perhaps some OOD generalization method should be applied to the OOD method selection problem, this concern does not outweigh the strenghts of the paper. Consequently, I recommend accepting the paper.

*Strengths.* The question posed by the paper is interesting and relevant to the reliability community. The experiments span synthetics, image, and language datasets. The paper provides some insights into what the meta-learner learns, which is, in turn, insightful about the connection between dataset properties and OOD generalization methods. The reviewers also found the paper to be well-organized, easy to read, and clear. The reviewers also found the simple, intuitive, and extensible nature of the framework to be a notable strength.

*Weaknesses.* The selector learned may itself suffer from OOD-generalization. While the paper is empirical, there is room for more (informal) connections between the dataset properties and the theoretical underpinnings of the method. The paper only evaluates a limited number of OOD generalization algorithms. The inclusion of more state-of-the-art algorithms would substantially improve the paper. Specifically, another limitation of the paper is that it does not deal with mixtures of shift types. The algorithms to select from are also closely tied to one shift type (e.g., covariate shift and label shift). Perhaps the selection is much more complicated and less generalizable when more sophisticated algorithms that may address many shift types are included.

The author provided additional experiments, albeit limited, to show that learning on smaller-scale datasets still generalizes to larger-scale datasets---though only within the same orders of magnitude. They also provided additional error metrics and other empirical results requested by the reviewers. Their clarification of their contributions was generally convincing to the reviewers.

The strengths of this paper outweigh its weaknesses; this provides a potentially useful strategy for addressing the distribution shift. Therefore, I recommend accepting the paper.